# Drawings of real-world scenes during free recall reveal detailed object and spatial information in memory

Wilma A. Bainbridge[1], Elizabeth H. Hall[1] & Chris I. Baker [1]

Understanding the content of memory is essential to teasing apart its underlying mechanisms. While recognition tests have commonly been used to probe memory, it is difficult to establish what specific content is driving performance. Here, we instead focus on free recall of real-world scenes, and quantify the content of memory using a drawing task. Participants studied 30 scenes and, after a distractor task, drew as many images in as much detail as possible from memory. The resulting memory-based drawings were scored by thousands of online observers, revealing numerous objects, few memory intrusions, and precise spatial information. Further, we find that visual saliency and meaning maps can explain aspects of memory performance and observe no relationship between recall and recognition for individual images. Our findings show that not only is it possible to quantify the content of memory during free recall, but those memories contain detailed representations of our visual experiences.

[1] Laboratory of Brain and Cognition, National Institute of Mental Health, Bethesda, MD 20814, USA. Correspondence and requests for materials should be addressed to W.A.B. (email: wilma.bainbridge@nih.gov)

When we recall a previously experienced event, what exactly are we remembering? Are our memories a precise, high-definition recording of that event, a low-resolution gist of that memory, or even just a verbal description of what we saw? Answering this question is an essential component of being able to tease apart the mechanisms of memory: what information is encoded and maintained, how memory decays over time, and what information is retrieved from these memories. Here, we present a large-scale examination of the information content in visual memories.

Previous work has often tackled this question from the angle of visual recognition, or the ability to identify a previously seen item as familiar or not. Some studies have reported a high capacity in recognition memory for thousands of object images[1–3]. However, this high capacity memory may contain relatively low detail—with observers likely using image gist to determine recognition[4], thus making recognition memory prone to errors such as change blindness[3,5] and spatial errors[6]. It is also unclear what specific content (e.g., the whole image, specific objects, or idiosyncratic features) drives successful recognition of an image.

Alternatively, memory can be tested using free recall in the absence of any explicit cues or foil images. Such studies have generally used simple stimuli including single words[7,8], images with few, isolated objects[9,10], or line drawings[11–13]. However, for complex stimuli such as real-world scenes, free recall is challenging to measure. Prior work on free recall of complex stimuli has often employed verbal metrics, having participants encapsulate a visual memory into a single word[7,9,11,12] or brief verbal description[13–15], but these measures provide limited insight into the content within those memories. Such verbal task-based studies suggest that recall suffers from low capacity, with participants recalling on average fewer than nine items regardless of the number studied[16]. An alternative approach is to use drawing, which can be considered a visual recall task. Drawings have been used to understand mental schemas of familiar objects[17–21] and cognitive differences between artists and non-artists[22–24], and computational models are being developed to aid with drawing recognition and production[25,26]. The ability to copy line drawings has also been used to diagnose memory disorders[27–29] and spatial neglect[30], and scene drawings have been used to look at effectiveness of encoding strategies[31] and the extension of boundaries in memory[10,32]. However, largely due to the complexity and subjectivity of drawings, such studies have often used small stimulus sets with simple metrics of interest (e.g.,[24]), or subjective experimenter ratings (e.g.,[10,31,32]), without delving into the rich content within these drawings.

Although recognition and recall have both been used to probe memory, they may also not reflect the same underlying mechanisms. Recognition and recall show dissociable activation in the brain[33–35] and can be impaired separately by different lesions[36]. Thus, the detail uncovered from recall memory may be very different from what has been explored until now using recognition tasks.

In the current study, we present a multi-pronged exploration of the content within memory, using a drawing-based visual recall task with complex, real-world images. To objectively quantify these 2682 resulting drawings, we leverage online crowd-sourcing (on Amazon Mechanical Turk, AMT), and recruit thousands of blind scorers to assess drawing diagnosticity, number of objects, extraneous objects, spatial detail, and object size within the drawings. We also examine the degree to which different image-based metrics of perception can explain which objects are ultimately recalled, and compare metrics of recognition and recall. Ultimately, we reveal detailed object and spatial content within visual recall memory, with observers creating precise drawings from memory for novel scene images.

## Results

**Delayed free recall memory task.** We conducted four separate drawing experiments to assess the amount of information in recalled memories. First, we will discuss the results for drawings made after a delay (Delayed Recall), before comparing those results with drawings made immediately after study (Immediate Recall), directly from the images (Image Drawing), and from scene category names (Category Drawing) to establish the nature and strength of memory representations.

For Delayed Recall, participants ($N = 30$ participants, in-lab) studied 30 images each from a different scene category, for 10 s each, knowing they would be tested on their memory later but not knowing the nature of the test. To ensure enough variability in memory performance for the stimuli, a random half of the images they studied were chosen to be highly memorable (based on recognition performance in a previous large-scale memory experiment[37,38]), and the other half were chosen to have low memorability (counterbalanced across participants), however, participants were unaware of this manipulation. Next, participants performed a difficult 11-min digit span task meant to limit verbal maintenance of these items in working memory and to introduce a delay between study and test phases. Specifically, they viewed a sequence of randomly generated numbers of 3–9 digits in length, had to remember each sequence and then verbally repeat it[2]. Performance on this digit span task was found to have no correlation with performance on recall or recognition of the images in the experiment (Supplementary Note 5). In the Delayed Recall test phase, participants were then asked to draw as many images from the study phase as they could remember with as much detail as possible, and were given as much time as they needed to complete the task. To assess whether there was additional information contained within memories that could not be initially accessed, participants were then cued with a diagnostic object for each category (e.g., bed in a bedroom) and were allowed to draw a separate set of any new images they could not recall before. Finally, participants completed a recognition memory task (indicating old/new for images) in which they were presented with the 30 images they originally studied, randomly intermixed with 30 foil images from the same scene categories. We then leveraged online crowd-sourcing on AMT to score the memory drawings for multiple properties, including object, spatial, and size information.

**Drawings from memory are diagnostic of their original image.** The first key question is whether drawings made during recall were in fact representations of their original image, or were only gist-based representations of the scene category. To test this, AMT workers ($N = 24$ separate participants per image, 1101 overall) were asked to match each drawing to one of three images from the same scene category. Delayed Recall participants accurately recalled 12.1 images (out of 30) on average (SD = 4.0, min = 5, max = 20) (see Fig. 1a, for example memory drawings; see Supplementary Figure 1 for additional example drawings). For comparison, in a similar free recall task with verbal stimuli, participants on average recalled 16.7 scene category names out of 30 (SD = 5.7; see Supplementary Note 2). Importantly, the Delayed Recall drawings were correctly matched to their original images from among same category foils by 84.3% of AMT workers on average (SD = 10.9%). When provided with a cue of a diagnostic object in each scene, participants recalled an additional 5.7 images on average (SD = 2.9, min = 0, max = 12), which were matched correctly by 80.3% of AMT workers on average (SD = 20.7%). There was no significant difference in diagnosticity between the free and cued recall drawings (non-parametric two-tailed independent

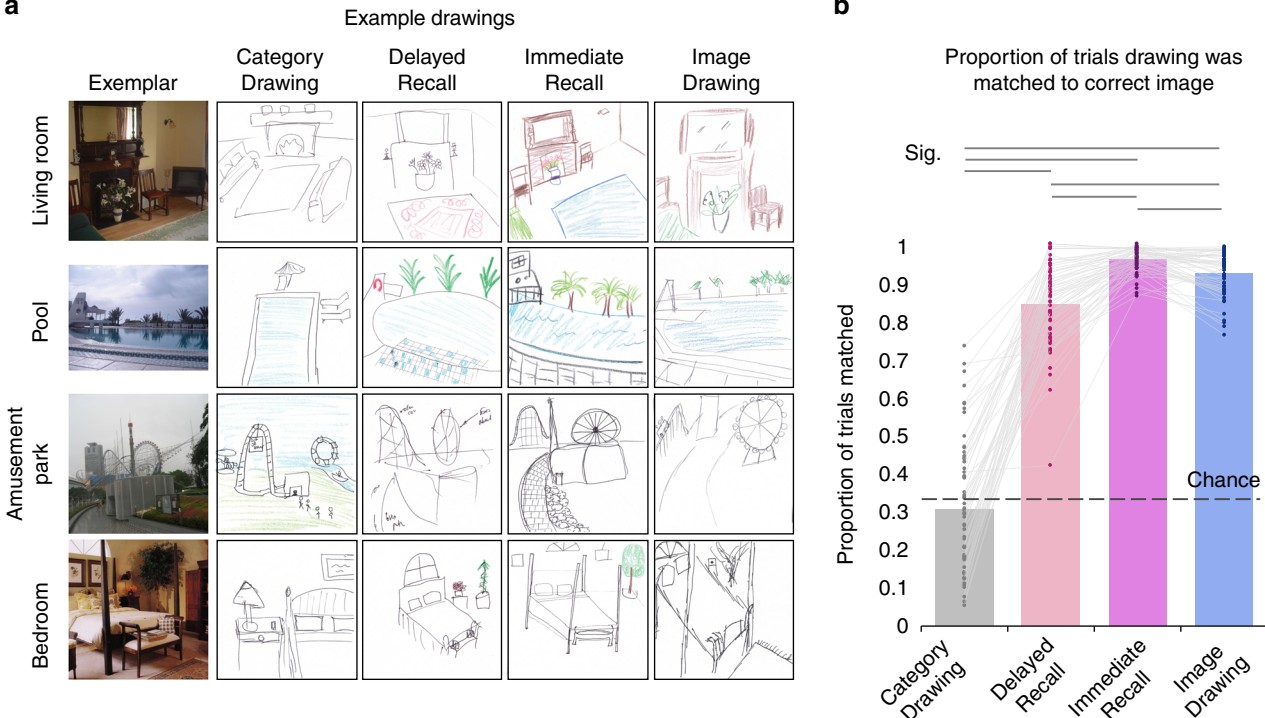

**Fig. 1** Example drawings and drawing matching performance. **a** Example drawings made from Category Drawing, Delayed Recall, Immediate Recall, and Image Drawing conditions for four exemplars from the 30 image categories (see Supplementary Figure 1 for examples from all 30 categories). Both Delayed Recall and Immediate Recall participants drew complex drawings, including multiple objects, spatial relationships of objects, and spatial layout of the scene. The Category Drawings show what sort of information is present in the canonical representation of each category. Delayed Recall and Immediate Recall participants are clearly using information from memory beyond just an image's category name, given the accurate object and spatial information in their drawings. **b** The average proportion of correct AMT worker matches of each drawing type (Category Drawing, Delayed Recall, Immediate Recall, Image Drawing). Each dot indicates each of the 60 images used in the experiment across drawings, and lines connect the same image across the different drawing conditions. Horizontal lines above the graph show all significant pairwise Wilcoxon rank-sum test comparisons at a Bonferroni-corrected level of $p < 0.0083$. The Category Drawings indicate the average proportion of matches with the two exemplars used in the study, even though there is no "correct" answer for this condition. All scene images in the manuscript are from the publicly available SUN Database for research of scene images[49]

samples Wilcoxon rank-sum test, WRST: $Z \sim 0$, $p \sim 1$), suggesting that participants may have more images within memory than those they are able to freely recall.

While these results suggest that participants maintained a specific representation of the scenes in memory, they do not establish the nature and strength of those representations and the extent to which they reflect the information encoded into memory or how memory can decay over a delay. To serve as critical benchmarks, we conducted three additional experiments with separate groups of participants (Figs. 1a, b, see Methods).

First, to estimate the extent to which the Delayed Recall drawings reflect memory for the scene category only and not the content of the images, one group of participants ($N = 15$ participants) was presented with the scene category names (e.g., "kitchen") and asked to draw corresponding pictures (Category Drawing). These Category Drawings reflect the individual participants' canonical representations of those scene categories, showing what and where objects would exist in their stereotypical version of a scene in the absence of memory for a specific image. If the images are equally representative of their category, then they should be matched around chance. However, given the variation in these real-world stimulus images, this condition also allows us to account for any biases in the stimuli or drawings of the stimuli. Although Category Drawings did show a spread in terms of matching frequency (Fig. 1b), they were on average matched near chance ($M = 30.7\%$ of AMT workers, SD = 17.4%). A Bonferroni-corrected ($p < 0.0083$) WRST on proportion of

AMT workers correctly matching each drawing revealed that Delayed Recall drawings were significantly better matched than Category Drawings ($Z = 9.29$, $p = 1.58 \times 10^{-20}$), showing that drawings from memory contained information beyond just a canonical category representation; these drawings were not merely constructed from a simple verbal label of the scene category, but contained additional visual information specific to the images.

Second, to estimate the maximum information one could draw from an image and to control for drawing ability, a second group of participants ($N = 24$ participants) drew directly from each image with no memory component (Image Drawing). 92.5% of AMT workers (SD = 5.8%) correctly matched Image Drawings to their corresponding image. Image Drawings were matched significantly better than Delayed Recall drawings (WRST: $Z = 4.49$, $p = 7.10 \times 10^{-6}$), showing that although Delayed Recall drawings were diagnostic of their image, they contain less information than a perceptual representation of an image.

Third, to determine what information is immediately encoded into memory and how this information decays over time, a third group of participants ($N = 30$ participants) made drawings 1 s after studying each corresponding image for 10 s (Immediate Recall). In all, 96.0% of AMT workers (SD = 3.2%) correctly matched Immediate Recall drawings to their corresponding image. Immediate Recall drawings were matched significantly better than Delayed Recall drawings (WRST: $Z = 6.78$, $p = 1.19 \times 10^{-11}$), showing that

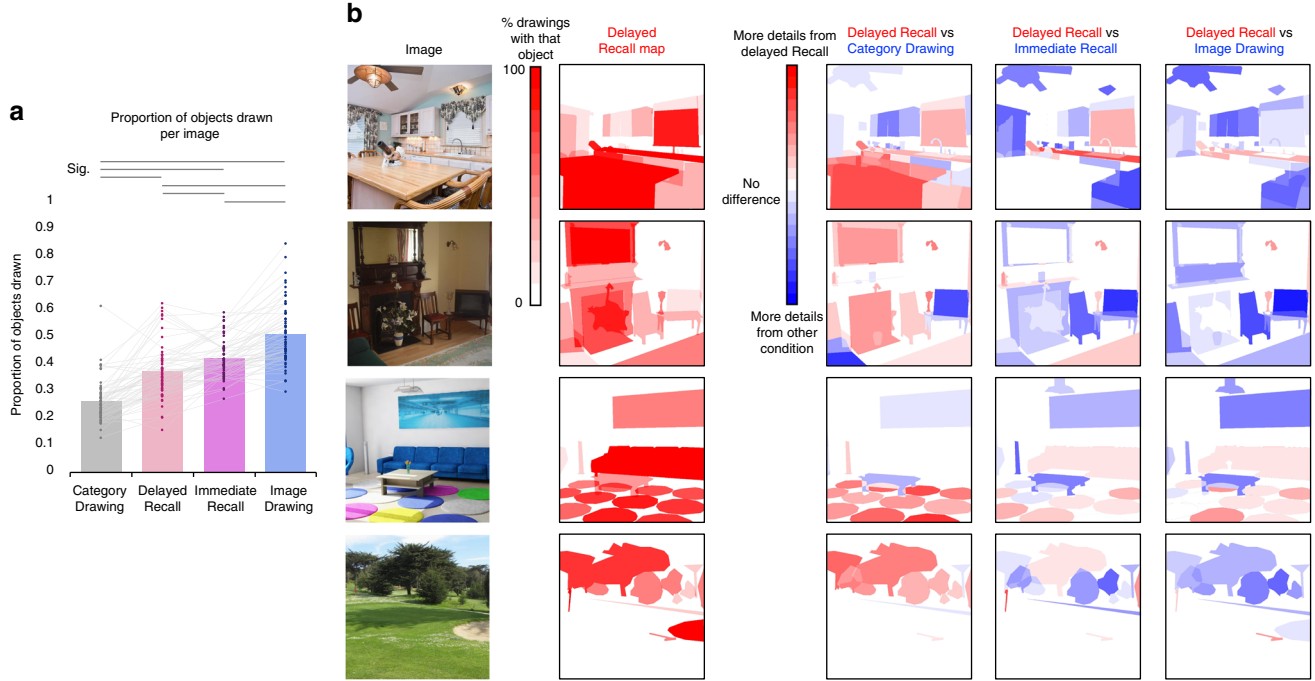

**Fig. 2** Comparison of objects drawn across conditions. **a** Average proportion of objects drawn for each drawing condition (Category Drawing, Delayed Recall, Immediate Recall, Image Drawing). Dots indicate average proportion for each of the 60 images used in the experiment, with lines connecting the same image across conditions. Horizontal lines above the graph indicate significant pairwise Wilcoxon rank-sum test comparisons that pass a Bonferroni-corrected significance level of $p < 0.0083$. **b** Example heatmaps of which objects were remembered. The "Delayed Recall Map" shows the drawing frequency of each object in the Delayed Recall drawings. Bright red indicates objects remembered by all participants who drew the image, and white indicates objects that were not remembered by anyone (white also indicates the background). The heatmaps on the right indicate the difference between the Delayed Recall heatmap (red) and the corresponding heatmaps for Category Drawing, Immediate Recall, and Image Drawing (blue), where white is a neutral color (background and objects that were drawn with equal frequency in both conditions). There were generally more objects in Image Drawings and Immediate Recall than the Delayed Recall drawings (e.g., more blue in the "Delayed Recall vs Image Drawing"), but there were also several objects participants remembered equally well (e.g., the flowers in the living room, the table in the kitchen), or even drew more frequently from memory than when perceiving the image (e.g., the hoe in the golf scene, the chef in the middle of the kitchen table). Image Drawings and Immediate Recall also show extremely similar heatmaps, showing the objects recalled immediately after encoding are much like those drawn at perception. The "Delayed Recall vs Category Drawing" heatmaps show that Delayed Recall drawings contained several items beyond what would exist in a canonical image from that scene category (e.g., circular rugs in a living room, a table in a kitchen), but there are also some objects that would be canonically drawn but participants did not successfully recall (e.g., the television in the living room with the fireplace, cupboards in a kitchen)

performance in Delayed Recall experienced some decay in memory over a delay. Permutation tests (100,000 iterations) confirm all of the above statistical tests (all $p < 10^{-5}$).

Collectively, these results indicate that participants were able to recall multiple images each, and their drawings from delayed recall were diagnostic of the original image, adding detail beyond a purely canonical reconstruction from memory of the category label. At the same time, these drawings still showed less diagnosticity than drawings made while viewing the image or immediately after encoding it, showing memory decay over time. However, while the drawings from memory are diagnostic of their image, this could reflect the presence of a small number of idiosyncratic objects or features rather than a detailed representation of the image. Further, the comparison of the memory drawings to the Category Drawings does not preclude the possibility that memories for these images may be stored using a verbal description of the image (see Supplementary Note 3 and Supplementary Figures 3–4). Although short verbal descriptions are diagnostic of their image, they also contain much less concrete information than is evident in the drawings. In the following sections, we characterize the detail in memory by quantifying the specific object and spatial information present in the drawings across the different conditions.

**Memory drawings contain numerous objects**. To characterize the extent of information in each drawing, we asked AMT workers ($N = 5$ participants per object per image, 2161 overall) to judge for each drawing which objects were present or absent, compared with the original photograph. Delayed Recall participants' drawings were not only diagnostic of the original image, they also contained multiple objects (Fig. 2a). Participants recalled on average 151.3 objects (SD = 55.1, min = 65, max = 282) across the experiment, or on average 11.4 objects per image they recalled (SD = 1.8, min = 7.5, max = 14.8). If one removes from this count any objects that were drawn by even a single participant for Category Drawings (i.e., objects that might exist in the canonical representation of a scene), the Delayed Recall participants still recalled on average 40.8 additional objects (SD = 16.7) across the experiment, or on average 3.6 additional objects per image (SD = 0.61).

Different categories contained different numbers of objects (on average $M = 19.2$, SD = 9.6, min = 5, max = 45) and if we analyze the data at the image level (Fig. 2), each Delayed Recall drawing contained on average 37.4% of the objects in each scene (SD = 10.1%), or an average of 7.0 objects (SD = 3.6). In contrast, Category Drawings contained on average 26.5% (SD = 7.8%) of the objects within the original images, meaning that essentially

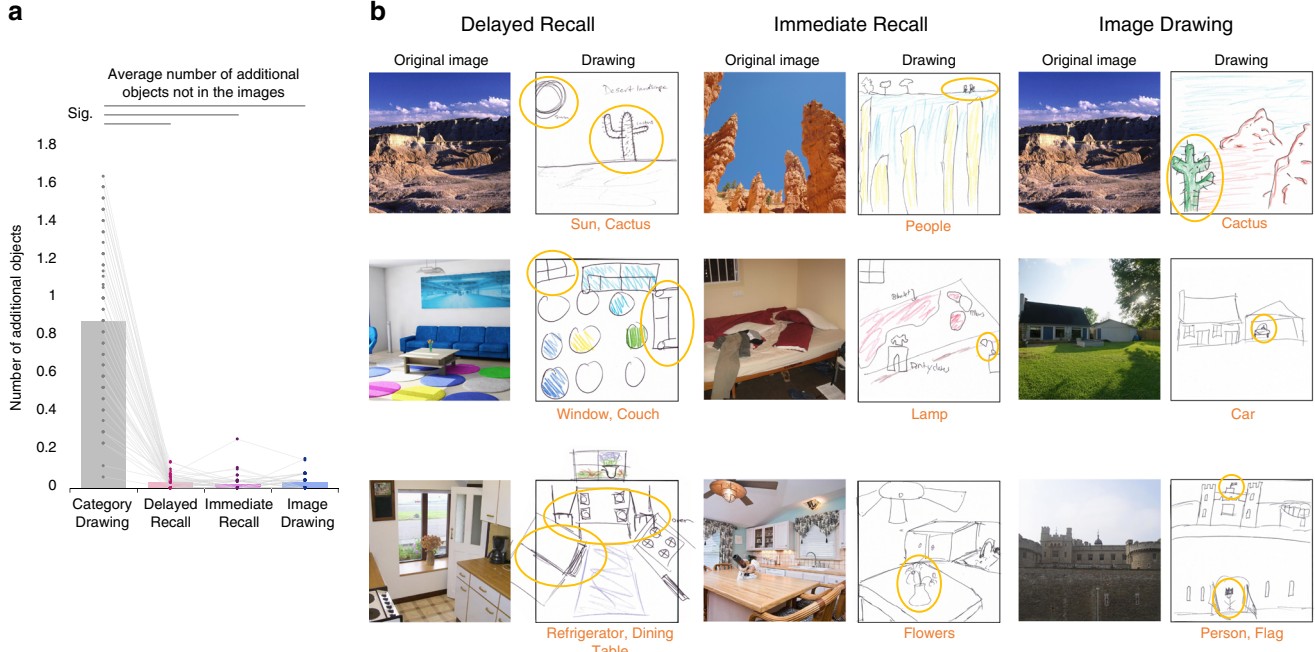

**Fig. 3** Comparison of additional objects drawn beyond those in the images across conditions. **a** Average number of objects drawn for each drawing type that did not exist in the images (Category Drawing, Delayed Recall, Immediate Recall, Image Drawing). Participants drew few additional objects when recalling images regardless of delay and when drawing from the image. For Category Drawing, they often drew objects that did not exist in the images for that label, as expected since drawings were merely done from the category name. Dots represent each of the 60 images used in the experiment, with lines connecting the same image across conditions. Horizontal lines above the graph indicate significant pairwise Wilcoxon rank-sum test comparisons that pass a Bonferroni-corrected threshold of $p < 0.0083$. **b** Examples of additionally drawn objects from Delayed Recall, Immediate Recall, and the Image Drawings. Additional objects are circled and labeled below each drawing in orange. Participants drew additional objects in the Delayed Recall drawings, for example, adding a cactus to a desert scene, drawing a window not captured in the image, or adding a dining table to a kitchen. However, participants also drew additional objects when recalling the image immediately after seeing it, adding people to a mountain scene, or replacing a chef sculpture with a vase of flowers in a kitchen scene. Even more surprising, participants drew additional, non-existent objects when drawing from the image itself, also adding a cactus to a desert scene or people and cars to scenes that did not have any

26.5% of the objects in a scene are objects that exist within the canonical version of that scene. Category Drawings contained significantly fewer objects than Delayed Recall drawings (WRST: $Z = 6.34$, $p = 2.37 \times 10^{-10}$). Image-based drawings contained many more objects, with on average 51.5% of objects (SD = 11.5%), or 9.4 objects (SD = 4.0) per image, and significantly more objects than Delayed Recall drawings ($Z = 6.46$, $p = 1.04 \times 10^{-10}$). Immediate Recall drawings contained on average 42.1% of objects (SD = 6.9%), or 7.8 objects (SD = 3.4) per image, and had significantly more objects than Delayed Recall drawings ($Z = 3.55$, $p = 3.84 \times 10^{-4}$), but fewer objects than Image Drawings ($Z = 4.97$, $p = 6.86 \times 10^{-7}$). All statistical results were confirmed with 100,000-iteration permutation tests.

In order to visualize the objects that were drawn in different conditions, we created object heatmaps that showed the percentage of participants who drew each object out of those who drew that image (Fig. 2b; object outlines were traced using tool LabelMe[39], see Methods). These heatmaps can then be subtracted between conditions to show which objects were drawn more frequently in one condition (e.g., Delayed Recall) versus another (e.g., Category Drawings). Although Delayed Recall drawings contained many objects beyond the Category Drawings, there were also some objects contained in the images that were drawn more frequently in a canonical version of the scene (i.e., Category Drawing) than from memory. For example, in Fig. 2b, cabinets in the kitchen, a television and table in a living room, and a road in a golf course were all drawn more often in Category Drawings than Delayed Recall drawings (see Supplementary Note 4 and Supplementary Figure 5 for a comparison of the

Category Drawings and the Delayed Recall Drawings at the object level). When compared with the Delayed Recall Drawings, the results for the Immediate Recall and Image Drawings are similar. For both, although there were several objects drawn in these conditions that were not recalled after a delay, there were also some objects drawn by Delayed Recall participants that were not included in the Immediate Recall or Image Drawings (See Supplementary Note 4 and Supplementary Figure 6 for a quantification). For example, in Fig. 2b, the hoe in the golf scene, the wine bottle in the chef's hand in the kitchen, the wall sconces in the first living room, and the blue couch in the second living room were all drawn more frequently in Delayed Recall than Immediate Recall and Image Drawings. These differences may highlight the importance of certain distinctive objects for recalling an image over a delay.

**Memory drawings contain few incorrect objects**. The quality of memory can also be assessed through the extent of inaccurate information. For example, perhaps participants in the memory conditions were drawing as many objects as possible based on the scene category, and so in spite of correctly recalling several objects, they were also drawing incorrect objects. We asked AMT workers ($N = 5$ participants per drawing, 817 overall) to judge for each drawing which objects were present in the drawing that were not in the image, or essentially the "false alarms (FAs)". For Delayed Recall, participants on average drew only 1.83 additional objects (SD = 1.97) from memory throughout the experiment (or on average 0.19 additional objects per image; Fig. 3), with

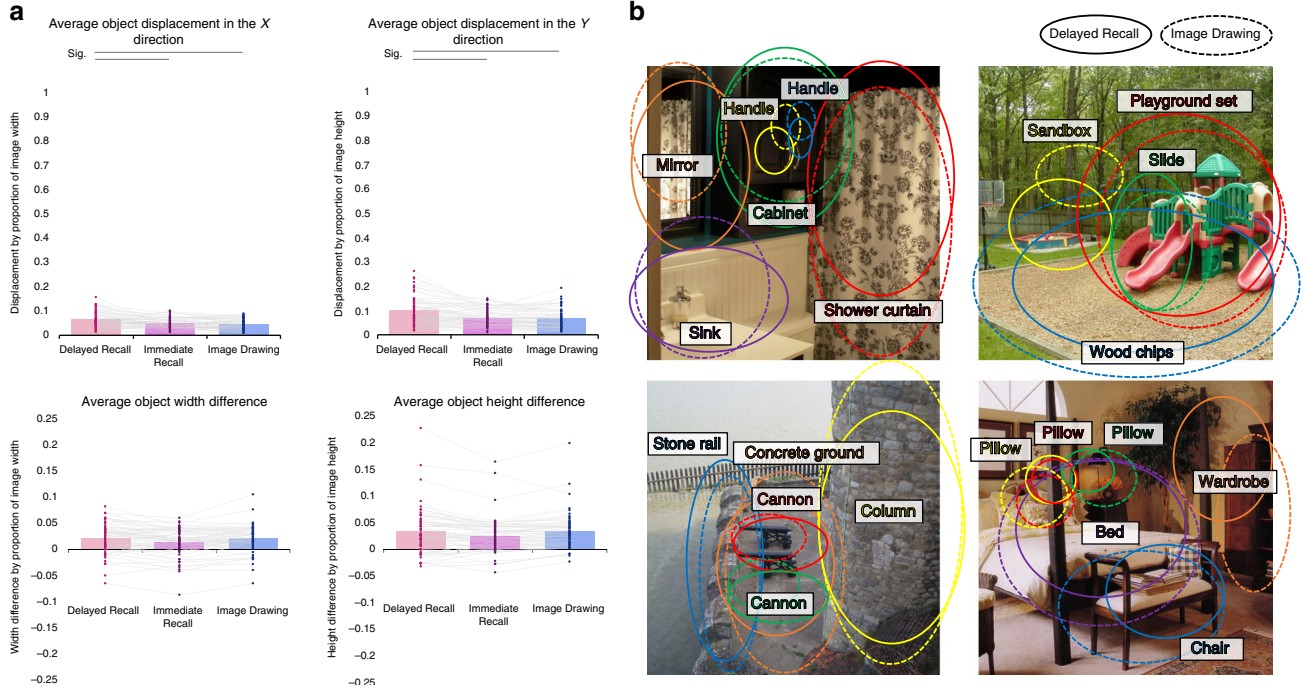

**Fig. 4** Comparison of object location and size across conditions. **a** Top—the mean X and Y distance between object centroids of the different drawing conditions (Delayed Recall, Immediate Recall, Image Drawing) and object centroids in the original image. Object centroids were determined from ellipses placed around each drawn object by AMT workers. The y-axes represent the distance as the proportion of the x-direction (or y-direction) pixel distance between centroids and the image pixel width (or height). Bottom—the mean ellipsis width and height differences between objects in each drawing condition and the objects in the original images. Y axis values represent width and height differences as a proportion of image width and height, respectively. Each dot represents each of the 60 images used in the experiments, and lines connect the same image across conditions. Significance in Wilcoxon rank-sum tests between conditions is indicated with horizontal lines at the top (Bonferroni-corrected $p < 0.0167$). **b** Example maps of the average ellipse encompassing the most commonly drawn objects in four of the images. Solid ellipses indicate the average object location in the Delayed Recall drawings, whereas dashed ellipses indicate the average object location in the Image Drawings. Participants in both conditions drew objects in the correct locations and at the right sizes; e.g., in the bathroom, putting the mirror in the upper left, the cabinet in the upper middle, the shower on the right, and the sink on the bottom left. This shows that participants drawing from memory had spatially accurate memory representations, drawing objects in the correct places and correct sizes from objects they had seen in images 11-min earlier

common additional objects including windows (six drawings), buildings (four drawings), the sun (four drawings), and trees (four drawings). As a comparison, the Category Drawings reveal the objects that exist in an individual observer's canonical representation of a scene category, beyond what is present in the specific image exemplars used in the study. This is a useful comparison, as it is possible that after drawing objects from memory, participants might fill in additional objects from that scene category type. Category Drawing participants ultimately drew on average 58.4 (SD = 24.7) additional objects across the experiment beyond those in the images used in the study. Delayed Recall drawings contained significantly fewer additional incorrect objects (WRST: $Z = 9.42$, $p = 4.24 \times 10^{-21}$), demonstrating that participants were not filling their recalled images with additional canonical objects. Surprisingly, Image Drawings contained on average 1.80 additional objects across the experiment (SD = 4.59; or 0.06 additional objects per image), drawing items such as people (seven drawings; no images in the experiment included people) and flags (four drawings), even though participants were viewing the image as they drew it. There was no significant difference between the number of additional objects in Image Drawings and Delayed Recall drawings ($Z = 0.22$, $p = 0.822$). Similarly, for Immediate Recall, participants drew on average 1.11 additional objects across the experiment (SD = 1.03; or 0.04 additional objects per image), with additional objects including windows or doors (four drawings) and extra circles (three drawings). Again, there was no significant difference

between the number of additional objects in Immediate Recall and Delayed Recall drawings ($Z = 2.50$, $p = 0.012$, did not pass Bonferroni-corrected threshold $p < 0.0083$). These results were replicated with permutation tests (100,000 iterations); significant results all $p < 10^{-15}$, whereas nonsignificant results all $p > 0.05$. In sum, these results show that participants' memories for objects are very accurate, and that they do not extend the contents of their memory much beyond the artistic license one might apply when drawing from an image directly.

**Memories are spatially accurate.** Although participants are able to recall many objects from the images with few additional objects, perhaps they remembered the set of objects within an image, but not the spatial layout. To assess the spatial precision of the memory drawings, we asked AMT workers ($N = 5$ participants per object, 4216 overall) to indicate the locations and size of each drawn object by creating an ellipse encircling each one. We compared the centroids and sizes of the ellipses across conditions with the true centroids and sizes in the original images calculated from ellipses around the object outlines (Fig. 4a).

In Delayed Recall, objects were placed extremely close to the original object locations, with their centroids on average displaced 6.5% of the image in the x-direction (SD = 3.2%) and 9.9% of the image in the y-direction (SD = 5.8%) from the centroids of the objects in the original image. In comparison, Image Drawings had 4.5% x-direction displacement (SD = 2.1%) and 6.6% y-direction displacement (SD = 4.1%), significantly closer to the true object

locations than objects in Delayed Recall drawings (WRST; $x$-direction: $Z = 3.58$, $p = 0.0003$; $y$-direction: $Z = 3.39$, $p = 0.0007$; Bonferroni corrected at $p < 0.0167$). Immediate Recall drawings also had low object displacement, with 4.7% $x$-direction displacement (SD = 2.1%) and 7.0% $y$-direction displacement (SD = 3.4%). Immediate Recall objects were placed significantly closer than Delayed Recall objects ($x$-direction: $Z = 3.21$, $p = 0.001$; $y$-direction: $Z = 2.68$, $p = 0.007$; Bonferroni corrected at $p < 0.0167$), and showed no difference from Image Drawings ($x$-direction: $Z = 0.55$, $p = 0.580$; $y$-direction: $Z = 1.07$, $p = 0.286$). These results were confirmed with 100,000-iteration permutation tests (all significant $p < 10^{-4}$). To determine whether the objects in Delayed Recall were placed closer to the original object locations than expected by chance, a permutation test was run by generating 100 randomly placed drawn object centroids per object per image, and calculating the average displacement over 1000 iterations. Objects in Delayed Recall were significantly closer to true object locations than chance ($x$-direction: $p < 0.001$, mean chance displacement: 30.9%; $y$-direction: $p < 0.001$; mean chance displacement: 30.3%).

We also tested the size of the objects, by comparing the heights and widths of the ellipses drawn around the objects in the drawings (Fig. 4a). In all conditions, the size of the drawn objects was close in both width and height to the original objects (calculated as difference between drawn object ellipse width/height and original object width/height, as a proportion of image width/height). In Delayed Recall, objects were on average 2.1% of the image wider (SD = 2.9%) and 3.4% of the image taller (SD = 4.5%). For Image Drawings, objects were on average 2.0% wider (SD = 2.6%) and 3.4% taller (SD = 3.4%). Immediate Recall objects were on average 1.3% wider (SD = 2.7%) and 2.6% taller (SD = 3.4%). Importantly, there were no significant differences in object size across the different drawing conditions (WRST: all $p > 0.1$), indicating that objects were drawn generally the same size regardless of memory condition.

Participants were thus not simply remembering the set of objects within an image or a sparse representation of object-to-object spatial relationships (e.g., "the wardrobe is somewhere to the right of the bed"), but were in fact preserving an accurate spatial map of the entire image.

**Image-based metrics can explain aspects of memory performance.** Given that certain objects were recalled more than others, what aspects of an object in an image might determine its likelihood of later recall? Understanding the features and objects in an image that drive recall may lead to an ability to better predict observer performance given an image, and to a greater understanding of how the brain codes information in memory. We thus tested the correlations between different state-of-the-art image-based metrics and which objects were ultimately recalled in both Delayed and Immediate Recall. We specifically tested two different algorithms shown to model image perception and be predictive of human fixation patterns: graph-based visual saliency (GBVS[40]), and Meaning Maps[41] (implementation of both models is described in the Methods). GBVS is a popular, biologically plausible visual model, meant to identify the most salient (i.e., visually dissimilar) regions in an image[38]. Such visually salient regions may ultimately be those that are strongest in memory and best recalled by participants in the current study. In contrast, the recently proposed Meaning Maps method posits that human attention is driven more by scene semantic content, rather than visual salience[41]. With this method, online participants rate the "meaningfulness" of patches from an image, to generate a semantic heatmap for this image. These Meaning Maps have been found to correlate more highly with human fixation patterns than

visual salience[42], showing that semantic meaning may guide human attention. It is thus possible that the components of an image that are ultimately recalled could be driven by semantic meaning within the image.

For all of the objects within images recalled by at least five participants (44 out of 60 images), we correlated the average saliency scores (ranging from 0 to 1) and the average meaning scores (ranging from 1 to 6) across the pixels of each object with the proportions of participants who remembered that object in both Delayed and Immediate Recall (Fig. 5a). All of the following results replicate regardless of whether object saliency or meaning is quantified as average score across object pixels, or peak score within an object.

For Delayed Recall, the probability of an object being recalled was significantly correlated with both object saliency (Spearman's rank correlation: $\varrho = 0.25$, $p = 1.27 \times 10^{-14}$) and object meaning ($\varrho = 0.19$, $p = 1.19 \times 10^{-9}$). When looking at the unique variance explained (using semi-partial correlations), saliency explained significant unique variance beyond meaning (Spearman's rank semi-partial correlation: $\varrho = 0.15$, $p = 3.07 \times 10^{-16}$), while meaning did not ($\varrho = -0.01$, $p \sim 1$). Although both of these metrics are correlated with the probability of an object being recalled, it could be that they are simply capturing the most diagnostic or canonical objects in a scene. For example, a bedroom is likely to contain a bed that might be visually salient or meaningful, but also present in any representation of a bedroom. If we account for the proportion of participants who drew the objects for Category Drawing, object saliency was still significantly correlated with probability of object recall ($\varrho = 0.07$, $p = 0.048$), but meaning was not ($\varrho = 0.05$, $p = 0.161$). Note that accounting for the objects in the Category Drawings is a particularly severe point of comparison, as this does not take into account the many additional objects beyond the images in this condition (compared with the low numbers in the recalled images).

For Immediate Recall, probability of object recall was significantly correlated with both object visual saliency ($\varrho = 0.24$, $p = 3.30 \times 10^{-14}$), and meaning ($\varrho = 0.24$, $p = 1.51 \times 10^{-13}$). Saliency explained unique variance beyond meaning ($\varrho = 0.08$, $p = 0.010$), and meaning also explained unique variance beyond saliency ($\varrho = 0.06$, $p = 0.050$). When taking the Category Drawings into account, saliency still significantly correlated with object Immediate Recall ($\varrho = 0.11$, $p = 0.001$), and meaning did as well ($\varrho = 0.13$, $p = 0.0001$). These results show that both visual saliency and meaning metrics of an image are able to capture some of the image features driving object recall, with some slight differences between the metrics and between Immediate and Delayed Recall.

However, much of the variance in the objects being recalled cannot be explained by these two models. To understand whether there were different general tendencies between these models and what is ultimately recalled, we visualized the average model-based maps, as well as the average object memory maps, normalized by the spatial distribution of objects across all images, i.e., dividing by the number of objects at each pixel location (Fig. 5b; see Methods). We find that both the saliency and meaning maps show a largely symmetrical distribution with a central bias. In contrast, the recall maps have an asymmetrical distribution with a tendency for relatively stronger recall of objects in the lower regions of an image, for both Immediate and Delayed Recall. These differences highlight that while such models may be sufficient to model fixation or attention patterns during perception, there may be unexplored spatial biases in the sorts of information that are ultimately recalled.

**Relationship between visual recognition and visual recall.** So far, the results show large object and spatial detail in visual recall

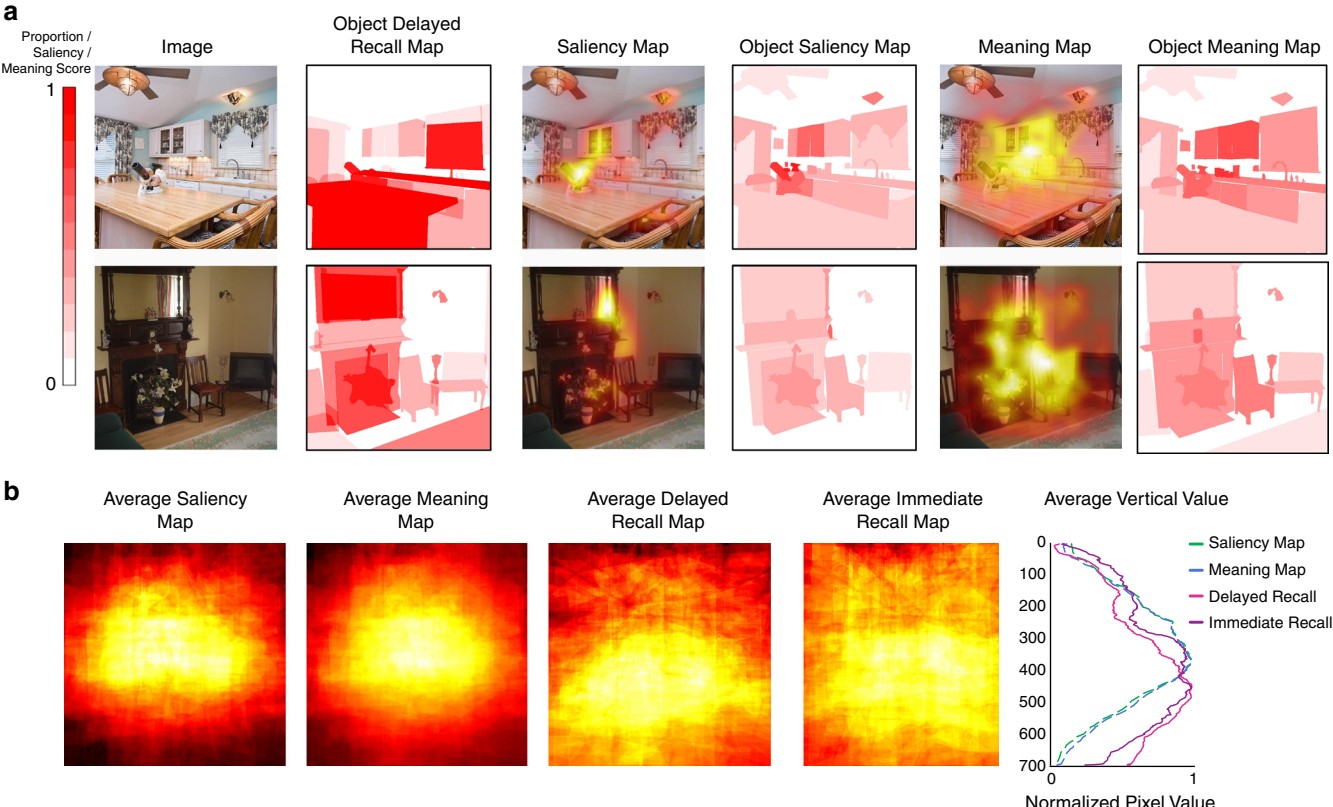

**Fig. 5** Comparing image-based metrics with object memory. **a** A comparison of which objects participants drew during Delayed Recall and the objects predicted by a graph-based visual saliency map (GBVS[40]) and Meaning Maps[41,42]. The Object Memory Map shows the proportion of participants who drew each object during Delayed Recall; red indicates objects drawn by all participants who remembered a given drawing, whereas white indicates objects drawn by no one, as well as background regions. The Saliency Map shows the saliency scores calculated for each pixel based on the GBVS algorithm, while the Meaning Map shows the average smoothed meaning scores attributed to circular patches taken from each image (see[41]). Their corresponding Object Maps show the average saliency and meaning scores across all pixels within a given object region, scaled to a range of 0 to 1. All results reported replicate when Object Maps are instead generated using peak saliency and meaning within an object. **b** The average maps across all experimental images, averaging (in order) the Object Saliency Maps, Object Meaning Maps, Object Delayed Recall Memory Maps, and Object Immediate Recall Memory Maps. Maps were normalized by number of objects across images at each pixel to take into account the natural spatial distribution of objects. The average vertical value shows the mean pixel value at each y-coordinate (from top pixel 0 to bottom pixel 700) for the four average maps, scaled to a range of 0 to 1. As one can see, while Saliency and Meaning Maps show a central bias, Delayed and Immediate Recall Maps show a tendency towards the lower part of the image

memory, reminiscent of the large capacity uncovered in visual recognition memory[3]. Do these results indicate two sides of the same coin, or might they reveal two separate memory processes? Recognition and recall show a neural dissociation in lesion studies[36] and neuroimaging work[35], and recognition has been reported to lack the spatial resolution and image detail we have shown here with recall[4,6], so we may expect to see differences in recognition and recall performance.

Prior studies have demonstrated that one can consistently measure an intrinsic "memorability" score for an image, based on performance in a large-scale online continuous recognition memory task[37,38,43]. However, does this recognition-based metric also translate to recall performance? To examine this question, we looked at how memorability of the stimuli (determined by prior large-scale online recognition memory tests) related to recognition performance, as well as recall performance. The images that participants studied comprised 30 high memorable images and 30 low memorable images, and we can thus compare their performance on both types of images. Overall, participants had very high recognition performance, with an average hit rate (HR) of 90.6% (SD = 8.72%) and an average FA rate of 10.2% (SD = 12.41%). As expected, image memorability was predictive of recognition task performance, with participants recognizing more high memorable images than low memorable images (paired

sample t-test: $t(29) = 4.11$, $p = 2.98 \times 10^{-5}$; low memorable: $M = 13.0$, SD = 1.49; high memorable: $M = 14.17$; SD = 1.56). However, low and high memorable images did not show a difference in recall performance; participants did not draw more high memorable images than low memorable images from memory ($t(29) = 1.61$, $p = 0.118$; low memorable: $M = 5.70$, SD = 2.32; high memorable: $M = 6.43$, SD = 2.37). That being said, a Bayes Factor analysis (Jeffrey–Zellner–Siow prior[44,45]) finds mild evidence in favor of the null hypothesis of 2.11:1, and numerically participants on average recalled more high memorable images than low memorable images.

However, when investigating correlations between recognition and recall, we find no significant Spearman's rank correlation between number of participants who recalled each image and the number who recognized it ($\varrho = -0.08$, $p = 0.541$), showing images that are more likely to be recognized are not more likely to be recalled. Similarly, there was no correlation between the number of images participants recalled and the number they recognized ($\varrho = 0.30$, $p = 0.109$), showing that at the subject level, recall ability is not necessarily linked to recognition ability. There was also no correlation between the number of objects drawn for a given image and its recognition rate ($\varrho = -0.02$, $p = 0.867$), showing that images that are more likely to be recognized also do not contain more detail in recall. Figure 6 shows the distribution

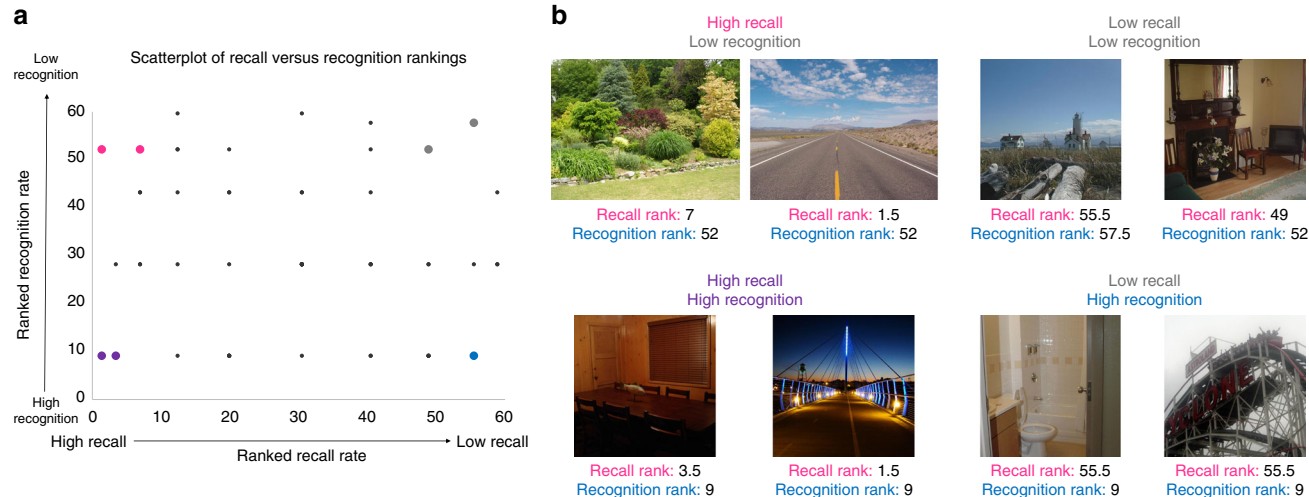

**Fig. 6** Comparing recognition and recall performance for images. **a** A scatterplot of the 60 images used in the experiment arranged by tied-ranked recall rate (proportion of Delayed Recall participants who successfully drew each image, using a tied rank where tied scores take on the average ranking) versus tied-ranked recognition rate (proportion of participants who recognized the image in the recognition task). Recall and recognition rates showed no significant Spearman correlation ($\varrho = -0.08$, $p = 0.541$), indicating there are different tendencies in the images one recalls versus those they recognize. **b** Example images that vary on the opposite ends of recall and recognition, determined by being beyond 1 SD above or below recognition and recall mean performance. Ranking number shows tied rank ranging from highest performance (1) to lowest (60). The points corresponding to these images are colored in **a**

of recall versus recognition performance for each image, as well as example images at each extreme (e.g., being one of the most recalled images, but least recognized). Collectively, these results are consistent with a possible dissociation between recall and recognition[33–36].

## Discussion

Drawings from memory reveal the object and spatial information maintained within visual recall memory after a delay. While one might expect the delayed recall task to be a very difficult task (drawing 30 complex, real-world scene images from memory after a taxing 11-min distractor task), participants recalled on average 12.1 images (and an additional 5.7 images when given a cue to unrecalled memories), with 151.3 objects total across those images. Importantly, participants remembered detailed information beyond a merely canonical representation of the scene category, drawing many distinctive objects (40.8 across the experiment) beyond the common objects predicted by the scene category. However, participants were not indiscriminately drawing all objects that might exist in a given scene, as their drawings were found to be incredibly precise, with few objects that were not present in the studied image. Further, participants' memories were spatially accurate, with objects drawn close to their original size and location in the image. Finally, this level of detail captures how information changes in memory over a delay, where some information diminishes (e.g., number of objects) in comparison with what is initially encoded, while other information persists (e.g., few additional objects, object size).

The current study presents a set of benchmarks of visual recall capacity and content, for both object and spatial information. While prior work has posited large capacity to visual recognition memory[1–3], other work has suggested recognition and recall memory may have surprisingly low detail[4,46,47]. Most work studying recall has also depended upon low-information verbal paradigms to estimate recall performance[7,8,11–14]. However, we demonstrate that drawings from memory contain meaningful and quantifiable knowledge, and any potential sources of noise (e.g., drawing ability) can be easily accounted for through additional comparison drawings (i.e., Category Drawings, Image Drawings,

and memory drawings after different delays) and rigorous large-scale online evaluation of these drawings. Beyond the main findings reported here, we are also able to replicate previously found memory phenomena, including boundary extension and primacy effects (see Supplementary Notes 6–7 and Supplementary Figures 7–8). This level of detail within our memories would be undiscoverable by previously used visual recognition paradigms, as they rely upon selecting matched foil images for the studied images, leading to assumptions about what aspects of an image are important components of a memory (e.g., spatial information, what FAs may occur). Our study also has important implications for clinical explorations of memory, which already frequently use drawing of abstract shapes to measure memory impairments (e.g., the Rey-Osterrieth Complex Figure Test[27,28]). Using similar methods to those explored in this study, these tests could be expanded toward more meaningful and complex images that can be scored more efficiently and objectively.

We also explored whether image-based metrics can explain what objects are ultimately recalled from an image. Both GBVS, which identifies visually dissimilar regions within an image, and Meaning Maps, which identifies meaningful patches within an image, were correlated with which objects people tend to draw over others. These correlations held even when accounting for those objects that would be drawn stereotypically from just the scene category name. However, these models also showed a strong central bias, whereas the objects recalled by participants tended to be in the lower parts of an image. Indeed, work on scene perception[48] posits that while scene category-based diagnostic information may exist centrally in a scene, more informative objects exist in the lower part of a scene. Thus, while saliency or meaning-based models reflect perceptual patterns, there will need to be further exploration on what features influence recall, and why such a lower region bias may exist. Such an exploration will also need to investigate how the canonical objects and their layout within a scene category serve as a prior to successfully or unsuccessfully drive attention or later recall. Direct measurement of eye movements during a similar memory task may reveal deeper insight into the relationship between the visual information attended to during encoding and what is drawn

during recall. Such information could ultimately guide us in the design of our visual world, so that we intentionally create scenes with objects we know we would likely recall. Additionally, crowd-sourced drawing and scoring of memory will help to enhance modern vision models (e.g.[25,26]) with the capacity to represent differences in content between perceptual, semantic, and memory representations of an image.

By comparing Immediate and Delayed Recall, we were able to assess the effect of memory decay. The diagnosticity of the drawings, number of objects drawn, and spatial precision all decreased with delay, whereas number of additionally drawn objects beyond the images and object size remained constant. Thus, different types of information within memory may be differentially affected by decay over time. Future work will be needed to investigate longer lengths of memory delay to understand the functional shape with which different types of information decay. Relatedly, one could be concerned that the increased delay and task fatigue could affect the quality of the images drawn later in time. While we do not see evidence of fatigue affecting recalled drawing quality in our current study (see Supplementary Note 1), the role of fatigue on memory performance is also an important question for future exploration. Also, while we do see a difference between Immediate and Delayed Recall, there are other differences between these two conditions other than the delay that might contribute to the effects we observe. Delayed Recall participants performed a mentally taxing task between study and recall, had to maintain all 30 images in memory concurrently, and did not know the nature of the recall task to be able to use an appropriate encoding strategy. In contrast, Immediate Recall participants only had to study a single image at a time, and underwent multiple study-test trials that could allow them to hone their encoding strategies to maximize drawing ability. Thus, additional work will be needed to tease apart how these different factors (strategy, memory load, memory delay, fatigue) influence ultimate recall performance.

The high level of detail within visual recall memory also highlights a separate aptitude from that previously found with visual recognition memory[3]. Although both types of memory may contain large amounts of information, the current study shows evidence for differences in what images are recognized versus recalled. Specifically, these two memory processes show differences in how they relate to memorability, an intrinsic image property reflecting the likelihood for an image to be later remembered[37,38,43]. Until now, memorability has only been studied using recognition tasks, and indeed in the current study, we replicate the effects of memorability—more memorable images were recognized better as being old (versus new) than less memorable images. However, for visual recall, we find that high memorable images are not necessarily more likely to be recalled than low memorable images, and there are no correlations between the images participants tend to recognize versus those they tend to recall. These results are consistent with prior studies reporting differences in the processes of visual recognition and recall. For example, in recognition participants easily confuse a mirror-flipped version of an image with the original[6]. In contrast, our recall results show precise spatial locations of objects, and no indication of mirror flips (the drawings show less $X$-displacement than $Y$-displacement). It is also debated whether there is in fact high object detail in visual recognition memory, as it is heavily dependent on the foil images and task used; high recognition rates are often reported only when two images are compared against each other in a two-alternative forced choice task, which may not require actual recognition, but just a difference in familiarity[4]. However, our recall experiment shows high detail even in the absence of any cues to drive a familiarity response. Thus, there may be important and meaningful behavioral

differences between visual recognition and recall. A next important step will be to investigate at a larger scale whether there are consistencies in the images and objects that are recalled across people (i.e., whether an image can be intrinsically recallable and if the recallability of a specific object drives recall of an overall scene). The task used in the current study could also provide a more standardized method to look at differences in recognition and recall memory performance in patients that have deficits in one process over the other (e.g., as in[36]).

There is a broad avenue of potential future research using drawings to measure visual free recall, and using online crowd-sourcing to score these drawings. A great amount of additional information can be measured about such drawings, for example, looking at specific object properties (color, rotation, occlusion, distance, object parts, etc), the spatial relationships between object pairs, and the errors people make in these measures. Although we find accurate object and spatial detail in people's drawings, they are still not pixel-perfect representations of the image, and so future work will need to explore what visual information is compressed (and in what way) versus what is accurately preserved, and what this can reveal about visual representations stored in memory. Relatedly, while the current experiment shows that memories for an image contain detail beyond basic category-specific information (with the Category Drawings), this does not fully rule out the possibility the observers could also be maintaining highly detailed verbal representations of these visual memories (although note the limited detail found in the Supplementary Note 3). Further experimentation will be needed to directly analyze and compare the amount of detail within verbal and visual representations of a scene, and see to what degree verbal and visual output modalities may differ or work in conjunction when recalling a memory. For example, if participants performed a verbal version of this task matched for difficulty and stimulus familiarity, what is the maximum amount of object and spatial detail that could be recalled? Additionally, work looking at the temporal dynamics of this recall process (e.g., order of drawing strokes over time) could contribute insight into how information is prioritized in memory, and how recollection develops over time. Finally, understanding the specific visual content of memories could guide important explorations into how representations of this content are stored in the brain and the neural mechanisms behind freely recalling an image.

Collectively, our findings show that something as seemingly unconstrained and variable as drawings can in fact be quantified to provide important insights into visual recall for real-world images. These drawings reveal that visually recalled memories can be accurate and detailed, while still prioritizing salient information.

## Methods

**Participants**. Thirty adults participated in the Delayed Recall experiment (21 female, average age 23.7 years, SD = 1.7), 30 different adults participated in the Immediate Recall experiment (20 female, average age 23.6 years, SD = 2.77), and 27 different adults (15 female, average age 24 years, SD = 4.14) participated in the Image Drawing experiment, Category Drawing experiment, and a Verbal Free Recall experiment. All participants were consented following the rules of the National Institutes of Health (NIH) Institutional Review Board and compensated for their time (NCT00001360). Additionally, 8596 participants were recruited on online crowd-sourcing platform AMT to score drawings from the study, following guidelines set by the NIH Office of Human Subjects Research Protections (OHSRP), and were also compensated for their time.

**Stimuli**. Ninety stimuli from 30 varied scene categories (three images per category) were used in the study. These images were taken from the Image Memorability Dataset[37] and the FIGRIM Dataset[38], using images and category names originally from the SUN database[49]. Categories were chosen to be as maximally different as possible, and included a mix of indoor (e.g., kitchen), natural outdoor (e.g.,

mountains), and manmade outdoor (e.g., street) categories (a list of all categories is in Supplementary Figure 1). Stimuli were sized so their longest side was 512 pixels and were shown to participants at approximately 14 degrees of visual angle.

The three images per category were chosen to contain a high memorable, a medium memorable, and a low memorable image, based on prior memorability data from an online memory test with thousands of participants[37,38]. Memorability here means HR (percentage of hits among repeats) in a continuous old/new task with on average 80 participants viewing each image. Memorability was used as an image selection metric in order to ensure there were no biases in memorability of the stimuli, and to see whether memorability could predict successful recall. High memorable and low memorable images were selected to be at least 0.5 SD away from the mean in the distribution found in Isola et al.[37], ($M = 67.5$, SD = 13.6). As a result, high memorable images had a mean HR of 85.1% (SD = 5.65, min = 75.0, max = 94.9), whereas low memorable images had a mean HR of 41.3% (SD = 9.6, min = 18.9, max = 53.6), both significantly different from each other ($t(58) = 21.54$, $p = 2.33 \times 10^{-29}$). Medium memorable images (used only as foils in the experiment) had a mean HR of 65.9% (SD = 6.39). Images were selected to have equally low FA rates, so that HR alone can be used as the main memorability metric (low memorable versus high memorable: $t(58) = 1.63$, $p = 0.11$; high memorable versus medium memorable: $t(58) = 0.21$, $p = 0.833$; low memorable versus medium memorable: $t(58) = 1.82$, $p = 0.074$). High memorable and low memorable stimuli showed no significant differences in low level visual features, including spatial frequency or color (R,G,B), tested with the Natural Image Statistical Toolbox[50]. Both high memorable and low memorable stimuli were also judged to be equally good exemplars of their scene category on a scale of 1 (poor example) to 5 (good example; low memorable: $M = 4.13$, SD = 0.66; high memorable: $M = 3.98$, SD = 0.70; t-test: $t(29) = 0.95$, $p = 0.351$; see Online scoring experiments in the Methods), showing that these images were well counterbalanced across conditions.

The outlines of the objects in each image were manually labeled by an experimenter using LabelMe[39], an online tool specifically for annotating the shapes of objects in a scene. Outlines were made before the experiments were conducted, and were created following the guidelines of[51]: objects were generally defined as nameable, separable, visually distinct items, larger than a 50-pixel diameter. Clusters of objects were often labeled together, especially if not visually distinct (e.g., dozens of trees in a forest). Visually similar object parts were not labeled separately from the object (e.g., leg of a chair), but large, detachable components were (e.g., knob on a stove). Although space-defining "objects" (e.g., wall, ceiling) were labeled, they were not included as objects in any of the experiments. Each image contained on average 21.1 objects (SD = 10.4, min = 6, max = 50). Example labels can be seen in the Supplementary Figure 2.

**Delayed Recall experiment**. The Delayed Recall experiment ($N = 30$ participants) consisted of five parts in order: (1) study phase, (2) digit span task, (3) free recall phase, (4) cued recall phase, and (5) recognition phase. This experiment, as well as the additional control experiments were conducted using MATLAB[52] and Psychtoolbox 3[53,54].

In the study phase, participants viewed 30 images, one per category, half of which were high memorable images and the other half low memorable images. Which category images were high memorable or low memorable was counterbalanced across participants so that each image was seen by 15 participants, and presentation order was randomized. Each image was presented for 10 s with an interstimulus interval (ISI) of 500 ms, for a total run time of 5 min. Participants were asked to study the images carefully as they would be tested on them later. However, details of the nature of the subsequent tests were not given (i.e., they did not know they would have to draw the images from memory).

The digit span task was used to ensure participants' memory for the images would not be based on verbal working memory, and to create a delay between study and test phases[2]. The verbal nature of the task should also decrease the likelihood that any recall effects we see are due to purely verbal encoding of the studied images, as this task should disrupt active verbal rehearsal. For this task, participants viewed a series of numbers presented sequentially (stimulus time = 1 s, ISI = 200 ms) and then were asked to verbally recite from memory the digits in order. The series of numbers ranged from three digits up to nine digits in length, to get a comprehensive measure of participants' verbal working memory. There were 42 digit series in total, resulting in a task time of approximately 11 min. Digit span task performance is reported in Supplementary Note 5.

During the free recall phase, participants were given a set of 30 blank squares at the same approximate size as the original images on sheets of paper and were asked to draw rough outlines for as many images as they remembered and then go back and fill in as much detail as possible. Participants were given as much time as they needed to draw and did not have to draw the images in order. Participants were provided a black ballpoint pen and colored pencils and were instructed to optionally color or label aspects of the image. Ultimately 56.3% of drawings contained color. Participants took on average 19 min total for this task, or approximately 1.7 min (SD = 0.8) per drawing.

In the cued recalled phase, for each studied image, participants were presented with a written cue of the most salient object in the image (e.g., a roller coaster in an amusement park, a toilet in a bathroom). Participants were then given the opportunity to draw a new image or add details to an already-drawn image, in a red pen (with the ability to still color and label the images). This phase is more

exploratory in nature, but can allow us to see if there were additional remaining memory traces of any images that participants were originally unable to access in the previous free recall phase.

Finally, in the recognition phase, participants completed a recognition task, where they were asked to indicate old/new and low/high confidence for a set of images that included all 30 images seen in the study phase randomly intermixed with 30 foil images of medium memorability from the same categories. Participants viewed each image until they responded, and after responding, each image was followed by an ISI of 500 ms.

**Additional experiments**. Four additional experiments were conducted, all with separate sets of participants from the Delayed Recall experiment. Participants in the first three experiments overlapped (participating in the order described to avoid confounding effects), whereas participants in the Immediate Recall experiment did not participate in any other additional experiments or the Delayed Recall experiment.

**Verbal Free Recall experiment**. A verbal version of the Delayed Recall experiment was conducted to serve as a loose comparison ($N = 15$ participants). Fifteen participants were recruited to match the 15 observations per image in the Delayed Recall Experiment. Participants studied the category labels (e.g., kitchen, amusement park) instead of specific images from the categories, using the same experimental timing as the study phase in the Delayed Recall experiment. They were then given the same digit span task as the Delayed Recall experiment. Finally, they were told to write down as many studied words as they could remember in any order (free recall phase). There was no cued recall phase for this experiment, as there were no objects within the words to cue, and there was no recognition phase as the recognition task in the Delayed Recall study involved foils of the same scene category. Results from this experiment are reported in Supplementary Note 2.

**Category Drawing experiment**. Fifteen participants were asked to draw images based on the category names used in the Delayed Recall experiment, presented in a random order. Fifteen participants were recruited to match the fifteen observations per image in the Delayed Recall Experiment. One participant stopped the experiment partway through (completing 10 out of 15 images). These drawings serve as a comparison with the Delayed Recall drawings in terms of what a canonical drawing or "gist" representation of a category would look like (e.g., there is always a bed in a bedroom). Specifically, this allows us to test if the images drawn from memory represent that specific image, or are just drawings based on solely remembering the category name.

**Image Drawing experiment**. Twenty-four participants were asked to draw a subset of the images used in the study phase of the Delayed Recall experiment, while looking directly at them. They were instructed simply to "draw this picture." Twelve participants per image were recruited (for 24 total), as no images from the Delayed Recall experiment were recalled by >12 participants. Each participant drew one image (randomly of either high or low memorability, counterbalanced across participants) per category (30 images). These drawings serve as a comparison with the Delayed Recall drawings in terms of what is the maximum detail one would naturally draw from any given image (e.g., you might not draw every book in a bookcase). This also serves as a control for what natural variations may arise based on people's differences in drawing ability.

**Immediate Recall experiment**. Thirty participants performed a recall task where they were asked to draw each image from memory immediately after viewing it, rather than after a delay. The same number of participants was recruited as the Delayed Recall experiment. Participants viewed each image from the Delayed Recall experiment for 10 s, and after a 1-s delay, were asked to draw the image they just viewed. They were given as much time as they needed, and instructions were identical to those of the Delayed Recall experiment. When they were done, they pressed a key to proceed onto the next image, and continued this task for 30 images. This experiment serves as a control for what information exists within memory immediately after image presentation, and to serve as a point of comparison for how information within memory may change after a delay and when items are no longer in working memory.

**Online scoring experiments**. Six online experiments were conducted on AMT to score the 2682 drawings that resulted from this study. A seventh online experiment was conducted to generate meaning maps for the images, as described in the Image-based models section. Online experiments were coded using HTML and JavaScript.

**Drawing matching online experiment**. AMT workers were asked to match each drawing to one of three photographs presented in random order: the low memorable, medium memorable (foil), or high memorable image from the same category as the drawing. In all, 1101 workers in total participated in this experiment, with 24 workers judging each drawing. Each worker could complete as many trials as they desired, and each worker on average completed 58.2 trials. This experiment

provides a measure of how specific each drawing is to its corresponding photograph, in comparison with the context provided by the category name. This experiment was also used to objectively score the results of the Delayed Recall experiment; only drawings where a majority of AMT workers agreed that the drawing matched its corresponding photograph (versus the opposite memorability photograph or medium memorability photograph of the same category) were scored as a successful recall trial. Note that the Category Drawings were not drawn from a specific image, and should thus be matched close to chance. Their performance was scored as the average of matching the drawing with each of the two image exemplars used in the Delayed Recall Experiment.

**Object marking online experiment**. This experiment was conducted to quantify which objects were drawn from each scene image. Workers on AMT selected, for each drawing, which objects in its corresponding image (indicated by an outline around each object) were contained in the drawing. In all, 2161 workers in total participated in this experiment, and five workers made a yes/no judgment for each possible object (in the image), for each drawing, completing on average 27.6 trials each. For a given drawing, an object was marked as being in the drawing if at least three workers agreed it was there. From this, one can create a heatmap of which objects were drawn, and the proportion of participants who drew each object (as in Fig. 2). The Category Drawings were assessed for the objects from both the low memorable and high memorable photographs of the same category.

**Additional objects online experiment**. AMT participants viewed each drawing and its corresponding image and were asked to write down the names of any additional objects in the drawing that were not in the photograph and if there were too many of an object (or "none" if there were not any). This provides a measure of the number of "false alarms" made (e.g., how many objects in the drawings were falsely remembered or drawn beyond what existed in the photograph). In total, 817 workers participated in this experiment, and five workers made judgments for each drawing, with workers completing 17.9 trials on average. An object was ultimately counted as an additional object if at least three out of five participants identified it. The Category Drawings were matched with both the low memorable and high memorable photographs of the same category.

**Object locations online experiment**. AMT participants viewed each drawing next to its corresponding image with an outlined object determined to be in that drawing (from the Object Marking Online Experiment), and were asked to place an ellipse encircling that same object in the drawing. Participants could both move and resize the ellipse. This experiment gets a measure of where each object was drawn in an image (from the centroid of the ellipse), as well as how large each object was drawn (from the ellipse height and width). In total, 4216 workers participated in this experiment, and five workers drew the ellipse for each drawing and object pair, each worker completing 21.9 trials on average. Ellipse coordinates were then transformed to be in the image-based coordinate system rather than the drawing-based coordinate system, and are reported based on proportion of the image height and width, as they varied across the experiment. This experiment was conducted for the Delayed Recall, Immediate Recall, and Image Drawings but not the Category Drawings, as there were few objects drawn in the Category Drawings that also existed in the original images.

**Boundary extension online experiment**. Previous work that has looked at drawings made from memory has found that memory drawings often contain larger boundaries than their original images[10]. To test this question for the drawings from the current study, AMT workers made ratings of boundary extension for the Delayed Recall, Immediate Recall, and Image Drawings. Specifically, participants viewed a drawing and its corresponding photograph and were asked to decide whether, "the drawing is closer, the same, or farther than the original photograph." They were also told to ignore any extra or missing objects in the drawing. Participants then responded on a five-item scale (as in[10]) ranging from –2 to 2 of much closer (–2), slightly closer (–1), the same distance (0), slightly farther (1), and much farther (2). They were also able to indicate can't tell if they were not able to judge (as sometimes drawings and images would be so different from each other that it would be impossible to compare distances). Seven AMT workers responded for each drawing, and a total of 301 workers participated in this experiment, with each worker completing 46.9 trials on average. This experiment was not conducted for the Category Drawings, as we anticipated the drawings would be too different from the photographs to be comparable, and it would be unclear what boundary extension or a lack of boundary extension would indicate for these drawings. Results from this experiment are reported in the Supplementary Note 7.

**Typicality online experiment**. Participants judged how representative each original image was of its scene category. This was to ensure that images used in the experiment were equally good examples of their category. Twenty-four AMT workers per image rated each image on a scale of 1 (poor example) to 5 (good example) for how representative it was of its category label. This task was modeled after the task in[55] used to assess scene typicality. Ultimately, 111 AMT workers participated in this experiment, with each worker completing 20.7 trials on average.

On average, images in the experiment were generally rated to be good examples of their category ($M = 4.06$ out of 5, SD = 0.68).

**Image-based models**. The GBVS model was implemented using the MATLAB toolbox by[40], using default parameters. The GBVS algorithm essentially identifies regions within an image that are highly visually dissimilar ("salient") from the rest of the image, and has been shown to significantly predict human fixations. The Meaning Map model was implemented as described in[41], and indicates regions of an image that are determined by observers to contain meaning. This has been shown to be predictive of human fixations, beyond the GBVS[41,42]. A seventh online experiment was conducted to generate meaning maps for the images.

**Meaning maps experiment**. Circular patches were cut out of the 60 images used in the experiment, at two different size levels scaled to match the same proportions of the images as used in[41]. This resulted in 16,921 small patches and 7829 large patches across the 60 images. On AMT, 1398 workers viewed 60 patches each, each from a different image and rated on a scale of 1 (low meaning) to 6 (high meaning) how "meaningful" each scene patch is. Three ratings were taken per patch and averaged to create a map for each of the two different sizes. These maps were smoothed with thin-plate spline interpolation, averaged between the two sizes, and then a center bias was introduced with a Gaussian blur scaled to the image size[41]. Both saliency and meaning maps were scaled so their range was from their minimum to maximum values.

To create the saliency-based and meaning map-based object maps, the average score was taken across the pixels contained within each object. Pixels belonging to each object were determined by the outlines on the original images created using LabelMe[39]. To create their average maps, the object maps for all 60 images used in the experiment were averaged and then scaled from 0 to 1. These maps were then normalized (divided) by a scaled (0 to 1) map of the number of objects at each pixel, in order to take into account the natural distribution of objects in an image.

**Code availability**. The code for these experiments are publicly available on The Dataverse Project at https://dataverse.harvard.edu/dataverse/drawingrecall and are linked to from the corresponding author's website.

## Data availability
The data for these experiments are publicly available on The Dataverse Project at https://dataverse.harvard.edu/dataverse/drawingrecall and are linked to from the corresponding author's website.

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

## Acknowledgements

We would like to thank Alex Martin and Adrian Gilmore for their valuable comments on the manuscript. We would like to thank John Henderson for his advice on implementing the Meaning Maps algorithm. We would also like to thank Wan Kwok and Alexis Kidder for help on the data collection. This research was supported by the Intramural Research Program of the National Institutes of Health (ZIA-MH-002909), under National Institute of Mental Health Clinical Study Protocol 93-M-1070 (NCT00001360).

## Author contributions

W.A.B., E.H.H., and C.I.B. conceived and planned the experiments. W.A.B. and E.H.H. conducted the experiments. W.A.B. analyzed the data. W.A.B., E.H.H., and C.I.B. wrote the manuscript.

## Additional information

**Competing interests:** The authors declare no competing interests.

