## [Peer Review File · Nature Communications]

Reviewers' comments:

Reviewer #1 (Remarks to the Author):

The manuscript is well written and interesting exploratory study of capacity of visual long-term memory with a tractable method. I like the method used and it is a nice way to quantify visual recall but this study while it supports what other studies have argued using different methods that we encode much more than just the gist of an image, it does not bring us any closer to understanding the mechanism by which this is done. We are no wiser as to how the brain does this. There are different bits and pieces that the authors draw from their data but the picture is fragmented and actually not very novel nor does it as I have said lead to any framework one could work with in understanding how one encodes or recalls visual data from long term memory. It tells us that we remember a lot of detail (this is not novel) but it is obvious that not all detail is made equal and this part is never explored. Because of this and other methodological concerns (mostly with the setting of benchmarks, see specific comments) that I feel I can not recommend for publication in this form.

1. If the authors would like to claim that massive visual long-term memory is very detailed than using only 30 images from 30 different semantic categories is not really probing that massive visual recognition memory.
2. Similar to the previous comment real-world images are helped by our knowledge of what the image should contain (e.g you know that a kitchen has a fridge) so the supposed incredible amount of detail is overstated.
3. Why use a digit span task if you want the observers not to rehearse? Why not give them a visual task instead to prevent maintenance in working memory?
4. The following finding which makes the bases of this paper: "For 95.6% of the memory drawings, a majority of AMT workers (N = 824) were able to successfully match that drawing to its correct original photograph from among same category foils." Does not give the real picture. This finding does not say much in favour of the claim that the images drawn from memory are very detailed since there is no baseline for comparison. If the same observers were asked to match a drawing with just one idiosyncratic feature or object of the picture to the original picture as a baseline (i.e. benchmark) then this finding would have meaning. As it stands subjects that matched the drawing to the original picture could have easily used only one diagnostic detail of object to guide their matches. One does not know. Using drawings from scene category labels as benchmark is not enough. It only takes care of effects of semantic knowledge one might apply. The upper bound benchmark while it is nice to have does not really tell us much because the lower band is not well defined.
5. How did the authors determine what is the ground truth, the number of objects in a scene to determine the percentage of objects reconstructed in the drawings? This is also the point that the authors in essence make: "In comparison, image-based drawings contained on average 51.5% of objects (SD = 11.5%), or 9.4 objects (SD = 4.0) per image. This serves as an upper bound of the maximum amount of detail one can expect when drawing a given image (e.g., no one will draw every flower in a garden);" Thus, it is not at the amount of detail that you provide but rather the detail itself

that you provide. It is what you drew not how much you drew. But the authors insist on giving us number of how much. We know from previous studies that there is a lot of detail we encode (not new) but can this data tell us anything about how we encode and what detail do we encode and reconstruct for others and ourselves to recognize?

6. I find that the recognition performance of the participants on only 30 images very low (e.g. only 50% even for highly memorable images) which is surprising, especially given the Konkel et al. findings and the fact that the subjects saw images for 10 seconds. The authors should address this.

Smaller points:

1. How did the author determine their sample size? Was there any power analysis done?

2. I have a problem with the images used since the two exemplars are really not at all prototypical of the category, what is more I would never categorize for example Exemplar 1 as an iceberg. It is actually a glacier. Same for Lake, exemplar 2, I would call this a pond and not a lake. Not using prototypical or mislabeling images means that you are intentionally undercutting your lower bound benchmark when you ask participants to draw prototypical category images. This is an issue here because you are using only 30 images and only 2 exemplars per category. [Other issue images Exemplar 1 for stadium is an ice rink; Exemplar 2 for Street is a road or highway rather than a street; Exemplar 2 for mountain is much more about the dirt road going up a hill than a mountain.

3. I understand the big number of AMT judges overwhelms the statistics but the fact that for each judgment there is a different number of judges and that the fact that not each AMT judge might give judgments for the whole set of 30. This is never reported and the big number of judges does not really tell us much. It is possible that each judge only judged one drawing and that is it, thus not developing any personal benchmark or criterion.

4. I believe that the comparison between recognized and recalled measure for low and high memorable images should not be based on the statistical significance but rather the effect sizes should be compared. So, for recognition a difference of approximately one image more recognized for high than low memorable images is significant whereas the same difference is not for recall. I think the effect sizes are more telling.

Reviewer #2 (Remarks to the Author):

This study investigates the issue of recall versus recognition in complex natural scenes. In an initial study phase, participants viewed 30 images (half highly memorable, and the other half low-memorability, determined from previous studies from first author) for 10 seconds each. Afterwards, the participants completed an 11-minute long digit span task. In the drawing phase, participants were given unlimited time to draw all scenes that were recalled in any order desired. After a cued-recall phase in which participants were given a salient item from each of the scenes to prime any additional memory traces, they performed an old-new recognition task in which each scene was to be chosen from a set of two images from the same scene category. The drawings created by these participants were then compared to a control group of participants who were either solely given the category label (lower-bound, testing for the canonical/prototypical nature of the scenes), or who drew the

images directly from the images (upper-bound, controlling for drawing ability). The drawings were then scored by online participants on Amazon's Mechanical Turk who matched the drawing to a foil, graded the correctness and completeness of objects' presence and location. While only 40% of scenes were drawn in the recall phase, this could be increased to 57% by cueing participants to a salient object in the scene. Of the drawn scenes, most were recognized by AMT workers as the correct scene, and the drawn scenes had a remarkable number of details and rather veridical spatial layout. The authors also found a surprisingly low false alarm rate for drawn objects. Additionally, they found that more salient objects were better remembered, even when controlling for the presence of canonical objects. Although participants were not more likely to draw a memorable picture, memorable pictures were better recalled in the final old-new task.

This is a provocative work with many intriguing tidbits. I commend the authors for being able to take such a seemingly unwieldy task (free drawing), and turn it into something scientifically quantifiable. I also feel that the upper- and lower- bounds, given by a different group of participants are clever. That said, I feel that to a certain extent, this work is over-hyped. I am also concerned about the "kitchen sink" nature of the study that seems to emerge over the course of the manuscript.

-1- The role of motivation versus memory

On average, only 12 scenes of the 30 (40%) were remembered during the drawing task, and no single participant came close to drawing all 30. I wonder what role fatigue, rather than memory per se, played a role in this. Several other results in the paper point to this as a possibility: participants did not draw more memorable versus less memorable scenes, and cued recall led to an almost 50% increase in the number of scenes drawn (tidbit that seems buried in the supplemental material) also points to a motivational factor. This is a critical point because the memorable vs non-memorable bit is the strongest bit of evidence that the authors provide for recall versus recognition, but it's not clear to me that this is about memory versus just wanting to get the experiment over with.

-2- Misleading and inappropriate quantitative metrics

-A- "For 95.6% of the memory drawings, a majority of AMT workers (N = 824) were able to successfully match that drawing to its correct original photograph from among same category foils." This is misleading because it leads the reader to believe that 824 people were rating a drawing, when in fact any given drawing was only being rated by N=24.

-B- "Participants remembered on average 151.3 objects (SD = 55.1, MIN = 65, MAX = 282) across the experiment.". The sheer number of objects remembered is not as important as the proportion of objects remembered. Scene categories vary widely in the number of objects within them. This measurement also interacts with the number of scenes drawn, and there was high variability here as well.

-C- It takes the authors until the 16th page of the manuscript to tell the reader that half of the 30 scenes were memorable, and half were less memorable. While this "big reveal" can be effective showmanship in a talk, it did not add to the clarity of the manuscript, and may be lost by a more casual reader.

-3- "Kitchen sink"

In the methods, the reader gets to see that this study had a number of potential avenues, only a small number of which were dealt with completely: cued recall versus free recall, digit span versus verbal memory, boundary extension, etc. Additionally, in the methods, the authors write about three levels of memorability (high, medium, and low), but throughout the manuscript, it appears as only two levels were used. While I appreciate the honesty of the authors in describing all experimental manipulations, I am left feeling that we just went on a very interesting fishing expedition.

-4- Data visualization

Figures that compare memory with upper- and lower- bounds: I am not sure why the authors chose a line graph representation of these categorical data. As there is no dimension linking any of the 30 categories to one another, this seems odd. I would recommend bar graphs. If the authors were using the line graph to show the overall level, they could consider dotted lines showing the mean for each condition.

Reviewer #3 (Remarks to the Author):

Summary

In this paper, the authors ask whether a drawing task can be used to measure the amount and quality of visual memory for visual scenes. They assess how well other, naive participants can match drawings made from memory to the original image cue, how many objects in the original image are included in the drawing, and how closely their locations in the drawing match their locations in the original image. They find that drawings produced from memory contain more information about the original image than if participants had not seen any image and only produced a drawing based on a verbal cue, and less information about the original image than if participants produced their drawing while viewing the original image. The authors also test how well a visual saliency algorithm predicts which objects are included in memory-based drawings, finding a modest correlation ($r=0.10$) between the saliency score for each object and the proportion of people including that object in their drawing. They conclude that drawing is a useful tool for measuring the content of visual memory.

Strengths

This paper has several major strengths: the question of what information is stored in visual memory is highly important, the main experiment and controls are well reasoned and well powered, and the results are clear. The figures make it very clear that the differences they measure between drawing conditions are systematic and reliable across scene categories. The use of crowdsourcing tools like Amazon MTurk made it possible to perform thorough and rigorous quantification of drawing behavior. The use of a saliency algorithm to predict what objects are included in memory-based drawings was appropriate and well-motivated.

Weaknesses

Meanwhile, the paper also raised several concerns for me regarding its suitability for publication in this outlet in its present form.

First, the interpretation of the memory-based drawing results based on relative distance from the label-only control vs. the image-based conditions is not well justified. The rationale for the label-only control is to provide a 'lower-bound' measure of the correspondence between participants' drawings and the original image. Of course, in the absence of image information, it is not surprising that such drawings do not contain as much image-level information as drawings produced of a specific image (whether from memory or from direct observation). However, the authors go further to claim on page 6 that "memory drawings were significantly closer to the upper bound than the lower bound ($Z = 8.89$, $p = 5.90 \times 10^{-19}$)." How was this statistic computed? Under what assumptions is it valid to conclude that being closer to the upper bound than the lower bound means that memory-based drawings are "largely diagnostic" or "only slightly less diagnostic than a drawing made directly from the image" (page 6), when this is based on a relative comparison to a weak baseline condition in which participants had no image-level information? In other words, even if memory-based drawings are more similar to the image-based drawings than the label-only drawings on the basis of containing more image-level information, this is not surprising because the label-only drawings were not produced with any image-level information.

Suggestions: If authors had wanted to make a claim about how well detailed image-level information is preserved in visual long-term memory, at risk of potential interference with memories for other scene images, a tighter control would have been a working-memory condition in which participants viewed each image for the same amount of time, and then immediately produced a drawing of it. Such a condition would have provided detailed information about what information was immediately encoded into visual memory, as well as the decay rate of such information in memory, thereby licensing more direct comparison with the long-term memory and image-based conditions. A variant of the label-only condition that would have been more comparable to the other two conditions would be to provide a sentence-length caption for the original scene image (also collected through crowdsourcing) that contains image-level information, better approximating the high-level perceptual/semantic representation of the scene image at encoding time. If the point is that visual memories contain richer detail than what can be compressed into a verbal encoding scheme of the objects and their spatial relations, etc., this would not only have been a tighter control for distinguishing the contribution of specific, detailed information about the image, per se, and the endurance of this information in visual long-term memory.

Second, at several points in the paper the authors claim that their observations about the content in memory-based drawings is "surprising." What licenses this impression? For the reasons above, that memory-based drawing fidelity is intermediate between the image-based and label-only conditions does not seem particularly surprising. Insofar as a reader has a strong prior belief that memory-based drawings should not contain any more image-level information than a category label would provide, then this result might be surprising, but it is not clear whether most readers would have this prior belief. So clearing that lower-bound baseline provides little constraint on our theoretical understanding of which

information is encoded into visual memory, such that it can be read out later in recall.

What are we supposed to think about the proportion of images that participants made drawings of: $12.1/30 = 0.403$? Could the authors comment on what this tells us about either the constraints on visual long-term memory (whether on encoding, maintenance, or retrieval processes), or constraints based on the visual recall task itself (increasing memory interference, amount of time to produce all drawings effectively increasing the delay period)? Did the authors note how long it took for participants to finish each drawing, all drawings?

The statistics reported on the probability of a drawing being matched to the original image (page 5) did not seem to be the most direct measures of this behavior -- rather than computing the proportion of images for which a *majority* of AMT workers (or any other single threshold) could identify the original image cue, why not just report the proportion of correct identifications, which makes it natural to report uncertainty on that estimate?

It is reported that memory-based drawings contained 37.4% of the objects in the original image. This means that the proportion of objects recalled is about $0.374 * 0.403 = 0.151$ of all objects in all original images, capturing what the authors estimate to be around 73.9% of the objects that are drawn when people aim to produce a drawing of image they can directly see. What the 73.9% statistic obscures is that for the 17.9 images that were not recalled, 0% of the objects were drawn, leading to approximately 15.1% of all objects being remembered (without the benefits of additional cueing).

On the other hand, for an image that a person has direct perceptual access to, the probability that they draw it should be close to 1, and so our best estimate of how many objects would be included is 51.5% of all objects (page 7). If so, then wouldn't a more appropriate way of estimating the proportion of objects recalled in the memory drawings relative to the image-based drawings be $0.151/0.515 = 0.293$?

Regardless, from either measure should we conclude that this reveals a "surprisingly high level of detail in memory?" Is there a principled choice of threshold that one of these measures could reach and be considered surprisingly high? E.g., is approx. 40% of all images "high"? Is ~30% of objects that would have been drawn "high"? What do we learn about memory for below-threshold images that were not drawn? Does this mean that participants had poor memory for them, or no memory at all?

Third, while the use of the visual saliency algorithm to predict what objects were included in memory-based drawings was well motivated and appropriate, GBVS appeared to only very weakly predict the probability that an object would be included in a memory drawing ($r=0.10$ after controlling for objects drawn in the label-only condition). This, to my mind, is the central theoretical opportunity provided by the visual recall paradigm in this study, and not adequately addressed in this paper. Are there alternative encoding models for predicting object-level memorability other than the saliency algorithm used? The authors cite Henderson and Hayes (2017) -- is there a good reason not to compare the results of GBVS vs. this meaning map algorithm? Could the authors comment on any relationship between

the data generated in their study and further development of such models (e.g., as training data)? I was unable to find additional details on the implementation of GBVS in this paper in the Methods section. How did the authors aggregate saliency scores across pixels within an object, when these diverged from one another?

Fourth, interrogating the contents of perceptual representations (including memory) with drawing tasks is hardly new, although the paper frames this approach as if it were, except for brief mention of work on boundary extension. Perhaps one of the most classic and iconic antecedents is Bartlett's experiments on serial reproduction of images from memory, which led to longstanding interest in how recall tasks can uncover subtle encoding biases, and/or lead to more schematized memory representations abstracted from the original sensory input. It might be useful to contextualize this work with respect to this theoretical backdrop, so the contribution of the current research can be more readily appreciated. Other work that may be relevant to review (and respond to) includes developmental studies on differential tendencies to draw what one knows ("intellectual realism," semantic knowledge) vs. what one sees ("visual realism," perceptual fidelity to source) across development -- e.g., studies by Paul Light, Gavin Bremner, and perhaps also Freeman & Janikoun (1972), Kosslyn et al. (1977). In addition, it might be useful to refer to Cohen & Bennett (1997), which asks which stages of processing most constrain accurate observational drawing (perception, decision-making, motor control, drawing evaluation). More recent psychophysical work of interest includes Florian Perdreau & Patrick Cavanagh's recent papers on effects of expertise on object encoding accuracy, as well as the work of Rebecca Chamberlain (w/ McManus and Wagemans).

Fifth, the stated contribution of this paper is methodological in nature (drawings can be used to measure visual long-term memory), rather than theoretical (using drawings to measure memory told us X,Y,Z new things about memory). In the introduction the authors suggest that such visual recall tasks could tell us something different from what we could learn from other visual tasks, e.g. recognition. Could the authors more explicitly state what we have learned about visual memory (rather than about the method) from using this visual recall paradigm? Without this, the theoretical significance of this paper is unclear. There are two ways that I believe that its significance could be substantively enhanced: (1) by including an additional memory-drawing condition that manipulates the delay either by shortening it (see Suggestions above) or extending it relative to the current main experiment, providing direct measurement of how the content and organization of information in visual long-term memory is affected by the relative demands on encoding vs. maintenance. (2) The other main way would be to strengthen the modeling section, motivating the use of different candidate visual encoding algorithms with respect to specific hypotheses about the role of perceptual, attentional mechanisms on constraining inputs to visual long-term memory, and comparing these different models' performance as a way of teasing apart these different hypotheses.

Minor: Could the authors comment on the decisions about drawing tools available, including the use of colored ink, in their task? It appears that several (but not all) drawings made use of color, but I was not able to find mention of this being evaluated systematically.

Minor: The measurement of spatial error used is limited by its reliance on x,y displacement on the picture plane, but this measure is insensitive to errors involving rotation, scaling, and occlusion. Could the authors comment on the potential limitations of the current method for measuring spatial reconstruction error?

The reviewers' comments have helped us make substantial improvements to the manuscript, which we feel have increased the impact and interpretability of our findings. Specifically, we have:

1. **Conducted a new experiment in-lab (N=30) investigating the content in memory after no delay (“Immediate Recall”)**. These 900 new drawings were scored with our series of online scoring experiments (N=2,929 new scorers). The Results, Discussion, and Methodology have all been modified to incorporate this new experiment. (Reviewer 3)
2. **Tested a new model – the Meaning Maps model** (Henderson & Hayes, 2017). This required an additional new online experiment to collect the meaning ratings for all image patches (N=1,398). (Reviewer 3)
3. **Conducted more testing of the GBVS and Meaning Maps models**. Specifically, we have tested alternate metrics for computing object maps and examined the distribution of memory and model-derived estimates in image space (Figure 5). (Reviewer 3)
4. **Investigated the possible role of fatigue on memory performance**, by analyzing how object and spatial detail decays over the course of the recall session. (Reviewer 2)
5. **Investigated object size in memory**, adding new analyses to see whether object height and width in memory drawings differ from their actual sizes (Reviewer 3).
6. **Added new analyses comparing recognition and recall**, including a new figure more clearly showing a lack of correlation between them (Figure 6). (Reviewer 1)

In addition, we have completely rewritten the Introduction, several parts of the Discussion, remade all figures, and conducted several additional smaller analyses and statistical tests as suggested by the reviewers. We think these substantial changes have greatly improved the manuscript.

Reviewers' comments:

Reviewer #1 (Remarks to the Author):

The manuscript is well written and interesting exploratory study of capacity of visual long-term memory with a tractable method. I like the method used and it is a nice way to quantify visual recall but this study while it supports what other studies have argued using different methods that we encode much more than just the gist of an image, it does not bring us any closer to understanding the mechanism by which this is done. We are no wiser as to how the brain does this. There are different bits and pieces that the authors draw from their data but the picture is fragmented and actually not very novel nor does it as I have said lead to any framework one could work with in understanding how one encodes or recalls visual data from long term memory. It tells us that we remember a lot of detail (this is not novel) but it is obvious that not all detail is made equal and this part is never explored. Because of this and other methodological concerns (mostly with the setting of benchmarks, see specific comments) that I feel I can not recommend for publication in this form.

Thank you for your thoughtful comments about the manuscript. We acknowledge that the current manuscript does not in itself reveal the mechanisms by which detailed visual information is recalled, but it does provide critical steps toward that goal. For example, to tease apart the neural mechanisms of memory recall, it is essential to understand what actual *content* and *detail* exists within this memory. Interpreting brain data acquired during recall is extremely difficult without clear and quantifiable metrics of the information recalled. Throughout the manuscript we have

clarified the goal of our work, and highlight how our findings provide new insight into the content of memory during free recall. We have also added new analyses that may provide some insight into the mechanisms of memory: **we have conducted a new Immediate Recall experiment** (changes throughout text) to see what types of information in memory decays over time, and **we also more fully describe a cued recall phase** (lines 103-106, 140-147) in the main text where we uncover higher numbers of detailed images in memory not initially accessible during free recall.

1. If the authors would like to claim that massive visual long-term memory is very detailed than using only 30 images from 30 different semantic categories is not really probing that massive visual recognition memory.

To clarify, we are exploring the issue of *detail*, not *capacity*. The latter would include being able to remember the gist of thousands of items, while the former would include remembering highly accurate and precise information at the level of single images. We are interested in the aspects of an image that are remembered, rather than familiarity for an image which might possibly rely on very limited information. Prior work has provided limited insight, due to the use of low-information verbal tasks or recognition tasks that require choosing foils based on assumptions of what detail to expect. While some prior studies have used drawing, this has typically been for low-information images (e.g., isolated object images), has focused on simple metrics of interest, or used subjective experimenter ratings.

Further, we want to highlight that our current work concerns visual *recall* memory, not visual *recognition* memory. Prior studies have demonstrated neural and behavioral dissociations of recall and recognition (e.g., Staresina & Davachi, 2006; Barbeau et al., 2011). While visual recognition may have large capacity (Brady et al., 2008), it may in fact have low spatial accuracy (Standing, 1970), and the level of actual detail is unclear (Cunningham et al., 2015). In contrast, our results clearly demonstrate that recall evidences high object and spatial information. Further, we find no correlation between the likelihood of recall and recognition across images (see new Figure 6).

We have highlighted these issues throughout the revised manuscript and hope the importance of our work is now clearer.

2. Similar to the previous comment real-world images are helped by our knowledge of what the image should contain (e.g you know that a kitchen has a fridge) so the supposed incredible amount of detail is overstated.

The reviewer highlights an important issue. It was precisely for this reason that we had a separate group of participants draw a picture when provided with just the category name (e.g., kitchen). This provides an estimate of what a drawn picture might contain given just semantic knowledge (e.g., most kitchens have fridges). In all our analyses, we contrast our memory conditions (Delayed Recall, Immediate Recall) with this baseline condition (Category Drawing). For example, on average, participants recalled 40.8 additional objects across the experiment beyond canonical objects that would exist in each scene category (e.g., a fridge in a kitchen), determined by drawings made from just the category name. Following the reviewer's example, it's interesting to note that neither of our kitchen exemplars contained a fridge, and no one drew a

fridge from memory. The only drawings with fridges were when people drew from the category name of kitchen.

3. Why use a digit span task if you want the observers not to rehearse? Why not give them a visual task instead to prevent maintenance in working memory?

We included the digit span task to decrease the possibility that participants were using a purely verbal strategy to remember the images (e.g., remembering the category names or a list of objects). However, as the digit span task requires visually seeing numbers, maintaining them in working memory, and repeating them, we anticipate the task should use a mixture of both verbal and visual working memory resources. This has been clarified in lines 99-100.

4. The following finding which makes the bases of this paper: "For 95.6% of the memory drawings, a majority of AMT workers (N = 824) were able to successfully match that drawing to its correct original photograph from among same category foils." Does not give the real picture. This finding does not say much in favour of the claim that the images drawn from memory are very detailed since there is no baseline for comparison. If the same observers were asked to match a drawing with just one idiosyncratic feature or object of the picture to the original picture as a baseline (i.e. benchmark) then this finding would have meaning. As it stands subjects that matched the drawing to the original picture could have easily used only one diagnostic detail of object to guide their matches. One does not know. Using drawings from scene category labels as benchmark is not enough. It only takes care of effects of semantic knowledge one might apply. The upper bound benchmark while it is nice to have does not really tell us much because the lower band is not well defined.

Measuring the ability of AMT workers to identify the original photograph from among same category foils was only our first analysis. It is precisely because this matching could potentially be achieved by focus on one idiosyncratic feature that we went on to quantify the number of objects remembered and their spatial location, revealing that participants remembered 11.4 objects on average per image with high spatial precision. Further, in the revised manuscript we have also found high accuracy for the size of objects in drawings made from memory. Collectively, these analyses demonstrate that recall of the images goes well beyond individual idiosyncratic features.

5. How did the authors determine what is the ground truth, the number of objects in a scene to determine the percentage of objects reconstructed in the drawings?

We determined the ground truth number of objects using a tool called LabelMe (Oliva & Torralba, 2001; Russell et al., 2007), which allows labeling of the outlines of objects within a scene image (lines 222-223, 773-774). These were labeled for the scenes by one experimenter (WAB), prior to the experiment. We have clarified these details in the manuscript.

This is also the point that the authors in essence make: "In comparison, image-based drawings contained on average 51.5% of objects (SD = 11.5%), or 9.4 objects (SD = 4.0) per image. This serves as an upper bound of the maximum amount of detail one can expect when drawing a given image (e.g., no one will draw every flower in a garden);" Thus, it is not at the amount of detail that you provide but rather the detail itself that you provide. It is what you drew not how much you drew. But the authors insist on giving us number of how much.

We know from previous studies that there is a lot of detail we encode (not new) but can this data tell us anything about how we encode and what detail do we encode and reconstruct for others and ourselves to recognize?

We now realize that that the term ‘detail’ might have been confusing. We operationalize detail as: number of objects, number of falsely recalled objects, spatial precision of objects, and size precision of objects. For the current study, we are not looking at detail within objects (e.g., object parts, textures); instead, we are looking at detail within scene images. These data provide extensive information about what information we encode and recall – based on prior work in visual recognition or verbal recall, we might have expected more intrusions (false memories) (Deese, 1959; Loftus, 2005), and low spatial precision (Standing, 1970; Cunningham et al., 2015). Throughout the manuscript, we have adjusted our language to avoid any ambiguity.

6. I find that the recognition performance of the participants on only 30 images very low (e.g. only 50% even for highly memorable images) which is surprising, especially given the Konkel et al. findings and the fact that the subjects saw images for 10 seconds. The authors should address this.

We apologize if this was unclear – *recognition* performance was in fact very high, and we have added a sentence in the text spelling out average recognition performance – an average hit rate of 90.6% and false alarm rate of 10.2% (lines 460-462). In contrast, participants freely *recalled* 12.1 items (40.3%) from memory, but with much object and spatial information.

Smaller points:

1. How did the author determine their sample size? Was there any power analysis done?

Since there has been no prior work using this type of paradigm on this scale, we unfortunately did not have data to use to perform a power analysis. However, we estimated that 30 participants should be sufficient to be able to capture meaningful effects in our data. Importantly, these data will now provide a basis for power analyses in similar future studies.

2. I have a problem with the images used since the two exemplars are really not at all prototypical of the category, what is more I would never categorize for example Exemplar 1 as an iceberg. It is actually a glacier. Same for Lake, exemplar 2, I would call this a pond and not a lake. Not using prototypical or mislabeling images means that you are intentionally undercutting your lower bound benchmark when you ask participants to draw prototypical category images. This is an issue here because you are using only 30 images and only 2 exemplars per category. [Other issue images Exemplar 1 for stadium is an ice rink; Exemplar 2 for Street is a road or highway rather than a street; Exemplar 2 for mountain is much more about the dirt road going up a hill than a mountain.

Scene category names (for 28 out of 30 categories) were taken from their categorization within the Scene Understanding database (Xiao et al., 2010), which has broad-reaching definitions of some scene categories – we have clarified this in lines 636-637. We also ran an online experiment asking AMT workers to rate how good of an exemplar each image was of its category name (based on Torralbo et al, 2013) on a scale of 1 (the image is a bad example) to 5 (the image is a good example). On average, the images scored 4.06 out of 5, with no significant

difference between the highly memorable and less memorable exemplars, indicating the images were generally considered good examples of their category (lines 661-666, 826-834).

3. I understand the big number of AMT judges overwhelms the statistics but the fact that for each judgment there is a different number of judges and that the fact that not each AMT judge might give judgments for the whole set of 30. This is never reported and the big number of judges does not really tell us much. It is possible that each judge only judged one drawing and that is it, thus not developing any personal benchmark or criterion.

We did have a set number of judges for each item for each drawing, and have now made this clearer on lines 132, 195, 270, and 316. **We have also calculated the average number of trials each judge performed for each experiment**, now reported in the Online Scoring Experiments section of the Methods (lines 765, 778-779, 790-791, 801-802), with each AMT worker performing 18-58 trials on average, depending on the task. As AMT judges did not know the nature of the drawings or the different conditions (i.e., that some were made from memory while others weren't), we anticipate any judging performance differences based on number of trials completed (e.g., developing a personal benchmark, or experiencing fatigue) should be balanced across drawing conditions.

4. I believe that the comparison between recognized and recalled measure for low and high memorable images should not be based on the statistical significance but rather the effect sizes should be compared. So, for recognition a difference of approximately one image more recognized for high than low memorable images is significant whereas the same difference is not for recall. I think the effect sizes are more telling.

This is a good point. **We have now investigated the relationship between recognition and recall performance in more detail and have added a new Figure 6** showing the distribution for recall versus recognition performance for all images. It is clear from this plot that there is no correlation between recognition and recall across images. Additionally, we have adopted more measured language when discussing the apparent dissociation between recall and recognition throughout the text to better reflect the data.

Reviewer #2 (Remarks to the Author):

This study investigates the issue of recall versus recognition in complex natural scenes. In an initial study phase, participants viewed 30 images (half highly memorable, and the other half low-memorability, determined from previous studies from first author) for 10 seconds each. Afterwards, the participants completed an 11-minute long digit span task. In the drawing phase, participants were given unlimited time to draw all scenes that were recalled in any order desired. After a cued-recall phase in which participants were given a salient item from each of the scenes to prime any additional memory traces, they performed an old-new recognition task in which each scene was to be chosen from a set of two images from the same scene category. The drawings created by these participants were then compared to a control group of participants who were either solely given the category label (lower-bound, testing for the canonical/prototypical nature of the scenes), or who drew the images directly from the images (upper-bound, controlling for drawing ability). The drawings were then scored by online participants on Amazon's Mechanical Turk who matched the drawing to a foil, graded the correctness and completeness of objects' presence and location. While only 40% of scenes were drawn in the recall phase, this could be increased to 57% by cueing participants to a salient

object in the scene. Of the drawn scenes, most were recognized by AMT workers as the correct scene, and the drawn scenes had a remarkable number of details and rather veridical spatial layout. The authors also found a surprisingly low false alarm rate for drawn objects. Additionally, they found that more salient objects were better remembered, even when controlling for the presence of canonical objects. Although participants were not more likely to draw a memorable picture, memorable pictures were better recalled in the final old-new task.

This is a provocative work with many intriguing tidbits. I commend the authors for being able to take such a seemingly unwieldy task (free drawing), and turn it into something scientifically quantifiable. I also feel that the upper- and lower- bounds, given by a different group of participants are clever. That said, I feel that to a certain extent, this work is over-hyped. I am also concerned about the “kitchen sink” nature of the study that seems to emerge over the course of the manuscript.

We appreciate the reviewer’s comments and have made numerous revisions to the manuscript to better characterize the significance of the work (and avoid the impression of over-hyping) and tighten the overall framework (and avoid the impression of the “kitchen sink”). We recognize that we have included many additional results in the Supplemental Information, but think that these help to characterize other aspects of free recall memory that we can extract and quantify from the drawings and that may be useful to researchers looking to explore this topic further.

-1- The role of motivation versus memory

On average, only 12 scenes of the 30 (40%) were remembered during the drawing task, and no single participant came close to drawing all 30. I wonder what role fatigue, rather than memory per se, played a role in this. Several other results in the paper point to this as a possibility: participants did not draw more memorable versus less memorable scenes, and cued recall led to an almost 50% increase in the number of scenes drawn (tidbit that seems buried in the supplemental material) also points to a motivational factor. This is a critical point because the memorable vs non-memorable bit is the strongest bit of evidence that the authors provide for recall versus recognition, but it’s not clear to me that this is about memory versus just wanting to get the experiment over with.

The question of motivation/fatigue is a particularly interesting point, and one that potentially affects any study of memory recall. There are three things we would like to highlight in response to this concern:

1. Motivated by your question, **we have conducted additional analyses to see if drawing detail decreases over the time of the task** (i.e., if the amount of detail drawn decreases the later an image is recalled) – see Supplemental Information. We found no significant correlation between recall order and number of objects drawn (mean Spearman’s $\rho = 0.007$, $p = 0.94$), nor with spatial displacement from the original locations of those objects (x-direction: mean $\rho = -0.01$, $p = 0.81$; y-direction: mean $\rho = 0.02$, $p = 0.73$), nor with a new analysis looking at the size of objects (width: mean $\rho = 0.04$, $p = 0.60$; height: mean Spearman’s $\rho = -0.08$; $p = 0.34$). These results suggest that drawing effort remained consistent over the task.
2. **The cued recall task occurred after the original free recall task**, so if fatigue plays a role, we might anticipate that the drawings obtained during cued recall would contain less information. However, since these cued memory drawings are matched by the online AMT workers in equally high proportion to the original images as the freely recalled

drawings (80.3% matched correctly), we think these drawings during cued recall argue against an effect of fatigue.

3. **The recognition task was conducted as the final task**, so we might expect the largest impact of fatigue here. However, participants show very high performance for recognition of the images, and still show that the intrinsic memorability of the image (determined from a separate online study) correlates with their recognition performance. In contrast, recall task performance, which was conducted earlier, shows no significant relationship to memorability.

We have added text in the Discussion (lines 556-560) on the potential role of fatigue in the experiment. We have also reorganized the Supplemental Information so information is less “buried”. Finally, we have added new discussion and a new Figure 6 further examining the relationship of recall and recognition.

-2- Misleading and inappropriate quantitative metrics

-A- "For 95.6% of the memory drawings, a majority of AMT workers (N = 824) were able to successfully match that drawing to its correct original photograph from among same category foils." This is misleading because it leads the reader to believe that 824 people were rating a drawing, when in fact any given drawing was only being rated by N=24.

We have added text (line 132) to make clear that each drawing was only rated by N=24, and have made similar changes for the other tests throughout the paper (lines 195, 270, 316). We have also added the average number of trials performed by each worker in the Methods (lines 765, 778-779, 790-791, 801-802).

-B- "Participants remembered on average 151.3 objects (SD = 55.1, MIN = 65, MAX = 282) across the experiment." The sheer number of objects remembered is not an important as the proportion of objects remembered. Scene categories vary widely in the number of objects within them. This measurement also interacts with the number of scenes drawn, and there was high variability here as well.

We have added the average number of objects drawn per image per participant and average proportion of objects drawn per image (lines 199-200, 207), as well as the average number of objects in the original images, standard deviation, and range (lines 205-206). We have also calculated a conservative metric of how many objects participants drew from memory beyond a canonical representation of that scene, by discounting any objects drawn from memory that were drawn by at least one participant from the category label: 40.8 objects across the experiment, or 3.6 objects per image (lines 200-204).

-C- It takes the authors until the 16th page of the manuscript to tell the reader that half of the 30 scenes were memorable, and half were less memorable. While this “big reveal” can be effective showmanship in a talk, it did not add to the clarity of the manuscript, and may be lost by a more casual reader.

We apologize – this was not intended to be a “big reveal”. We originally thought putting too much of the methodological details in the beginning might weigh down the paper, so we introduced each methodological detail in combination with the results related to them. **We have**

now added a description of the memorability of the stimuli early on (lines 91-96) to make this aspect of the design clearer.

-3- “Kitchen sink”

In the methods, the reader gets to see that this study had a number of potential avenues, only a small number of which were dealt with completely: cued recall versus free recall, digit span versus verbal memory, boundary extension, etc. Additionally, in the methods, the authors write about three levels of memorability (high, medium, and low), but throughout the manuscript, it appears as only two levels were used. While I appreciate the honesty of the authors in describing all experimental manipulations, I am left feeling that we just went on a very interesting fishing expedition.

We regret that we gave the impression of a fishing expedition. The primary goal of this study was to quantify the detail within recalled scene memory, and compare this performance to recognition performance. The Delayed Recall, Image Drawing, and Category Drawing experiments and methods were decided *a priori*. These are the details we focus on in the main text.

However, we also performed additional analyses post-hoc to see if our results could replicate classic results (e.g., boundary extension, primacy/recency) and to provide a full account of the data we collected (e.g. verbal recall, digit span). Although these results are not critical for our main conclusions, we think reporting them in some form (i.e., in the Supplemental Information) is useful and informative for other investigators wishing to pursue these questions in more depth the future. Further, we want to be transparent about all analyses we performed. However, if the reviewer thinks these analyses would be best left out, we would be willing to remove items from the Supplemental Information or move analyses from the main text to the Supplemental Information. We have organized the Supplemental Information to hopefully make its separate items clearer.

In terms of the three levels of memorability, the third level (medium memorability) is specific to the foils used in the recognition task and the drawing matching task for the AMT workers. During encoding, only low and high memorable images (two levels) were presented. We have now clarified this in the manuscript (line 654).

-4- Data visualization

Figures that compare memory with upper- and lower- bounds: I am not sure why the authors chose a line graph representation of these categorical data. As there is no dimension linking any of the 30 categories to one another, this seems odd. I would recommend bar graphs. If the authors were using the line graph to show the overall level, they could consider dotted lines showing the mean for each condition.

The reviewer makes a good point and **we have now redesigned all relevant Figures to show the data as bar graphs (Figures 1-4)**, and think the data are much clearer now. In order to still maintain some information about the individual categories, we have also included dots for each of the 60 images used in the experiment, with faint lines connecting same category images, so one can see within-category trends across the conditions. We hope this makes the data both easier to understand and also more transparent to the reader.

Reviewer #3 (Remarks to the Author):

Summary

In this paper, the authors ask whether a drawing task can be used to measure the amount and quality of visual memory for visual scenes. They assess how well other, naive participants can match drawings made from memory to the original image cue, how many objects in the original image are included in the drawing, and how closely their locations in the drawing match their locations in the original image. They find that drawings produced from memory contain more information about the original image than if participants had not seen any image and only produced a drawing based on a verbal cue, and less information about the original image than if participants produced their drawing while viewing the original image. The authors also test how well a visual saliency algorithm predicts which objects are included in memory-based drawings, finding a modest correlation ($r=0.10$) between the saliency score for each object and the proportion of people including that object in their drawing. They conclude that drawing is a useful tool for measuring the content of visual memory.

Strengths

This paper has several major strengths: the question of what information is stored in visual memory is highly important, the main experiment and controls are well reasoned and well powered, and the results are clear. The figures make it very clear that the differences they measure between drawing conditions are systematic and reliable across scene categories. The use of crowdsourcing tools like Amazon MTurk made it possible to perform thorough and rigorous quantification of drawing behavior. The use of a saliency algorithm to predict what objects are included in memory-based drawings was appropriate and well-motivated.

Thank you – we appreciate these comments.

Weaknesses

Meanwhile, the paper also raised several concerns for me regarding its suitability for publication in this outlet in its present form.

First, the interpretation of the memory-based drawing results based on relative distance from the label-only control vs. the image-based conditions is not well justified. The rationale for the label-only control is to provide a 'lower-bound' measure of the correspondence between participants' drawings and the original image. Of course, in the absence of image information, it is not surprising that such drawings do not contain as much image-level information as drawings produced of a specific image (whether from memory or from direct observation). However, the authors go further to claim on page 6 that "memory drawings were significantly closer to the upper bound than the lower bound ($Z = 8.89$, $p = 5.90 \times 10^{-19}$)." How was this statistic computed? Under what assumptions is it valid to conclude that being closer to the upper bound than the lower bound means that memory-based drawings are "largely diagnostic" or "only slightly less diagnostic than a drawing made directly from the image" (page 6), when this is based on a relative comparison to a weak baseline condition in which participants had no image-level information? In other words, even if memory-based drawings are more similar to the image-based drawings than the label-only drawings on the basis of containing more image-level information, this is not surprising because the label-only drawings were not produced with any image-level information.

The reviewer makes a very good point and **we have now removed this “significantly closer” statistic**, and modified language making inferences beyond what is statistically shown throughout the text.

Suggestions: If authors had wanted to make a claim about how well detailed image-level information is preserved in visual long-term memory, at risk of potential interference with memories for other scene images, a tighter control would have been a working-memory condition in which participants viewed each image for the same amount of time, and then immediately produced a drawing of it. Such a condition would have provided detailed information about what information was immediately encoded into visual memory, as well as the decay rate of such information in memory, thereby licensing more direct comparison with the long-term memory and image-based conditions.

Thank you for this excellent suggestion. **We have run a new Immediate Recall experiment** (working memory) with a new set of in-lab participants (N=30) and AMT scoring (N=2,929 overall). In this new experiment, participants viewed a given image for 10s, and then after a 1s delay, drew the image from memory. **We have modified the figures, results, methods, and discussion to account for this new condition.** We find that in some cases, delayed recall is indeed worse than immediate recall (e.g., in matching the drawing to the photograph, or number of objects), but in other cases they are the same (e.g., false alarms are equally low, object size remains constant). This shows that different aspects of memory recall may experience different amounts of decay over a delay, and serves as an interesting launching point for future work investigating longer delays.

A variant of the label-only condition that would have been more comparable to the other two conditions would be to provide a sentence-length caption for the original scene image (also collected through crowdsourcing) that contains image-level information, better approximating the high-level perceptual/semantic representation of the scene image at encoding time. If the point is that visual memories contain richer detail than what can be compressed into a verbal encoding scheme of the objects and their spatial relations, etc., this would not only have been a tighter control for distinguishing the contribution of specific, detailed information about the image, per se, and the endurance of this information in visual long-term memory.

This is a very interesting suggestion. We acknowledge that the category labels may not be a good proxy for a verbal description the participants could generate during encoding. To assess the feasibility of this suggestion, **we ran a pilot study asking a set of AMT workers (total N = 105) to “write a one sentence description of this image that you would use to remember it”** for all 60 images used in the experiment, with responses from 10 workers per image.

We found a wide variety of content in workers’ responses, with varying amounts of concrete object, concrete spatial, and subjective information. To roughly quantify these images, we had 3 naïve members of our lab rate each sentence for: 1) presence of the scene category name or a synonym, 2) number of concrete objects, 3) number of spatial descriptors for those objects, and 4) subjective detail, which includes qualitative adjectives and adverbs (e.g., beautiful, messy) as well as metaphors, emotions, and relations to personal experiences. The 10 responses for four images are shown below:

Key:

Black = Scene category

Red = Subjective detail: qualitative adjectives, metaphors, emotions

Blue = Concrete object detail

Green = Concrete spatial detail

1. We see an **unmade cot or bed** in a **plain room**.
2. An **unmade bed** with **white and burgundy sheets**, with **clothes** laying **on it**, in a **yellow room**.
3. A **very messy room** with **red bed sheets**.
4. **Someone needs to make this messy bed**.
5. **Laundry and cleaning day for sure**.
6. An **unmade bed** in a **small, messy bedroom**.
7. An **unmade bed**, covered in **red sheets**, **discarded clothing** and **various personal items strewn** on the **floor** beneath.
8. **Unmade bed** with **clothes** on it.
9. **Messy rooms because who seriously has time to make their bed every morning?**
10. **Messy bed** with **burgundy linens**.

1. A **beach resort pool** with **tile** and **palm trees**.
2. We see a **pool**, **perhaps at a hotel**, with **palm trees** behind.
3. The **pool** was **amazing** and **surrounded by palm trees**.
4. **Tropical paradise is right here!**
5. **When peace and quiet meet the sky, you will find me here.**
6. **My dream vacation pool.**
7. There is a **large pool** surrounded by **Middle Eastern tiling** with **palm trees** in the **background** of a **very sunny day**.
8. **Reflective pool in a tropical location.**
9. **Vacations would be so relaxing** in this **pool** with **gorgeous blue skies** all around.
10. The **pool**, the **tropical trees** and the **white building** remind of me a **time in Abu Dhabi**.

1. We see a **modern-looking kitchen** and, **out the window**, to the **street** beyond.
2. A **kitchen** with a **window** and a **door**, that has **white cabinets** and a **wood countertop**.
3. Welcome to this **inviting kitchen!**
4. The **best window seat in town**.
5. This is a **small kitchen**.
6. **Small white and wood kitchen.**
7. This **kitchen looks very lovely but kind of small**.
8. **Sad flowers** near a **window** of a **cramped kitchen**.
9. **Clean organized kitchen** that is cream and brown.
10. The **flowers** in the **vase** really **bring color** to this white **kitchen**.

1. This **garden** has a **walkway**, **red flowers** and a **spiral tree**.
2. This is an **English style garden** set in a **courtyard**.
3. We see a **landscaped courtyard**, with a **spiral-cut tree (a yew?)** in the **center**.
4. In a **garden reminiscent of Downton Abbey and old English royalty**, stands a **tall spiraling evergreen** surrounded by **red tulips**.
5. **Beautiful garden** walkway **maze**.
6. **That is a very nice looking garden.**
7. This **garden** is **beautiful as well as functional**.
8. The **garden area** with a **tree** in the **middle** and a **brick building** in the **background**.
9. This **beautiful green tree** surrounded by **flowers** and **shrubs** is **definitely the centerpiece**.
10. A **spiral tree** amongst **pink tulips**.

We found:

- 1) **These descriptions contain a low number of objects** ($M = 1.6$ objects per image, $SD = 0.8$) **and even lower number of spatial details** ($M = 0.6$ spatial words per image, $SD = 0.4$). These spatial details were also generally relations between two objects (“beautiful green tree surrounded by flowers and shrubs”), not the absolute spatial position in relation to the image. 90.2% of descriptions contained the scene category name or a synonym.
- 2) **There are high amounts of subjective detail** ($M = 1.5$ subjective adjectives and metaphors per image, $SD = 0.4$). This leads us to believe that many participants in this pilot task are creating personal cues that would not have meaning to someone else who might draw the image (e.g., “My dream vacation pool”, “...a garden reminiscent of Downton Abbey and old English royalty”, “Welcome to this inviting kitchen!”).
- 3) **There is a wide variety in people’s verbal strategies**. Some people only described the scene category type without object information, “That is a very nice looking garden”,

some wrote a list of objects, “This garden has a walkway, red flowers, and a spiral tree”, while others wrote personally meaningful descriptions, “Tropical paradise is right here!”.

Based on these preliminary findings, we think the question of what semantic representation someone might use to encode a memory is in fact very complex. It is likely to depend heavily on aspects like the instructions (here, we specified for them to write one sentence as the reviewer suggested, but what is the appropriate amount of information to ask for?), the task (do they form the sentence while viewing the image, or from memory?), or participants’ differing intuitions of what strategy to take for a verbal memory task (e.g., remember a list of objects versus link the image to something personally meaningful). There is also the important question of how to combine or select appropriate sentences to follow-up with a drawing study.

Because of this combination of factors, we think this question of an alternate “semantic lower bound” is beyond the scope of the current study, and better suited for a study directly comparing semantic and perceptual recall memory representations. We have clarified the wording throughout the manuscript to make clear that our Category Drawing condition is more to capture a canonical scene category representation (i.e., are participants solely remembering the scene category type?) and not a base semantic representation of the image – we cannot rule out that there is some form of verbal or semantic representation contributing to the recalled drawings.

Second, at several points in the paper the authors claim that their observations about the content in memory-based drawings is "surprising." What licenses this impression? For the reasons above, that memory-based drawing fidelity is intermediate between the image-based and label-only conditions does not seem particularly surprising. Insofar as a reader has a strong prior belief that memory-based drawings should not contain any more image-level information than a category label would provide, then this result might be surprising, but it is not clear whether most readers would have this prior belief. So clearing that lower-bound baseline provides little constraint on our theoretical understanding of which information is encoded into visual memory, such that it can be read out later in recall.

Fair point - we have changed our language throughout the text to limit such subjective claims on findings being “surprising”.

What are we supposed to think about the proportion of images that participants made drawings of: $12.1/30 = 0.403$? Could the authors comment on what this tells us about either the constraints on visual long-term memory (whether on encoding, maintenance, or retrieval processes), or constraints based on the visual recall task itself (increasing memory interference, amount of time to produce all drawings effectively increasing the delay period)? Did the authors note how long it took for participants to finish each drawing, all drawings?

The 40.3% recall performance at the image level is difficult to interpret reliably, as it is unclear whether it reflects limitations in capacity or something else, e.g., inability to retrieve an existing memory. However, there are two aspects of our data worth highlighting. **We have added a description of a cued recall phase (taken from the Supplemental Information)**, where, following the initial recall period, we cued participants with the names of single diagnostic objects within each image, and found participants could remember on average an additional 5.7 images, that were equally diagnostic as the originally recalled images (lines 140-147). This indicates that some of the constraints on visual recall memory may be inability to access encoded

memories, although this will need to be explored further. Second, **we have performed new analyses that reveal that number of objects drawn, spatial accuracy, and object size did not change for drawings made later during the experiment** (Supplemental Information). This provides evidence that fatigue or an increased delay to recall may not be a critical factor influencing recall performance and detail.

We did record how long it took participants to finish all drawings in the Delayed Recall experiment, but not how long it took to make each drawing. Participants took on average 19 minutes for the recall portion of the drawing task, an average of 1.7 min (SD = 0.80) per drawing (added to lines 697-698).

*The statistics reported on the probability of a drawing being matched to the original image (page 5) did not seem to be the most direct measures of this behavior -- rather than computing the proportion of images for which a *majority* of AMT workers (or any other single threshold) could identify the original image cue, why not just report the proportion of correct identifications, which makes it natural to report uncertainty on that estimate?*

We have changed the probabilities reported (lines 142-143, 160-162, 169-170, 177-178) to now instead report the proportion of correct identifications, including the standard deviation as a measure of variance on the estimate (capturing both worker uncertainty as well as response error).

*It is reported that memory-based drawings contained 37.4% of the objects in the original image. This means that the proportion of objects recalled is about $0.374 * 0.403 = 0.151$ of all objects in all original images, capturing what the authors estimate to be around 73.9% of the objects that are drawn when people aim to produce a drawing of image they can directly see. What the 73.9% statistic obscures is that for the 17.9 images that were not recalled, 0% of the objects were drawn, leading to approximately 15.1% of all objects being remembered (without the benefits of additional cueing).*

On the other hand, for an image that a person has direct perceptual access to, the probability that they draw it should be close to 1, and so our best estimate of how many objects would be included is 51.5% of all objects (page 7). If so, then wouldn't a more appropriate way of estimating the proportion of objects recalled in the memory drawings relative to the image-based drawings be $0.151/0.515 = 0.293$?

Our percentages report an image-based metric of the proportion of objects that are remembered on average for a given image when it is drawn (analogous to the heatmaps shown in Figure 2), so this metric does not take into account participants who did not draw this image. This is because we wanted to compare what information exists within the recall trace for these specific images across conditions. However, we also report subject-based metrics, including the number of images drawn by participants across the experiment, and have now added how many objects they drew per image remembered (lines 198-200). We have also added a metric of how many objects participants recalled in the Delayed Recall experiment beyond any objects drawn by the participants drawing from the category label to obtain a measure of unique objects in memory beyond a canonical scene representation (i.e., 40.8 objects across the experiment, or 3.6 objects per image; lines 200-204).

Regardless, from either measure should we conclude that this reveals a "surprisingly high level of detail in memory?" Is there a principled choice of threshold that one of these measures could reach and be considered surprisingly high? E.g., is approx. 40% of all images "high"? Is ~30% of objects that would have been drawn "high"? What do we learn about memory for below-threshold images that were not drawn? Does this mean that participants had poor memory for them, or no memory at all?

As these explorations of object and spatial detail within recalled visual images are new, indeed these measures of "high" are based against previous assumptions of the low quality or capacity of visual recognition or verbal recall (e.g., Miller, 1956; Levin & Simons, 1997; Loeftus, 2005; Cunningham et al., 2015). However, we have changed the text throughout our manuscript to temper our subjective wording. We hope that the current work serves as a set of benchmarks for future work quantifying the full bounds of visual recall capacity and detail.

The question about below-threshold images is very interesting and fairly complex. The addition of our cued recall results into the main text (lines 140-147) provides evidence that there are actually detailed visual memories that fail to be successfully recalled (possibly due to a failure to access that memory). This will be an important question for future exploration.

Third, while the use of the visual saliency algorithm to predict what objects were included in memory-based drawings was well motivated and appropriate, GBVS appeared to only very weakly predict the probability that an object would be included in a memory drawing ($r=0.10$ after controlling for objects drawn in the label-only condition). This, to my mind, is the central theoretical opportunity provided by the visual recall paradigm in this study, and not adequately addressed in this paper. Are there alternative encoding models for predicting object-level memorability other than the saliency algorithm used? The authors cite Henderson and Hayes (2017) -- is there a good reason not to compare the results of GBVS vs. this meaning map algorithm? Could the authors comment on any relationship between the data generated in their study and further development of such models (e.g., as training data)? I was unable to find additional details on the implementation of GBVS in this paper in the Methods section. How did the authors aggregate saliency scores across pixels within an object, when these diverged from one another?

At the suggestion of the reviewer, **we have now expanded this section and provided a more in-depth analysis where we apply both the GBVS and the Meaning Maps algorithm** (Henderson & Hayes, 2017; $N = 1398$ AMT workers participated) to these memory maps (for both immediate and delayed recall). We then look at amount of unique explained variance by these two models as well as by what objects are drawn in the canonical representation of that scene category (i.e., Category Drawings). We find both saliency and meaning are correlated with the objects that are ultimately recalled (with both Immediate and Delayed Recall). Both algorithms, however, do not capture a large proportion of the variance in recall memory, and when comparing the maps, we find a central bias in the saliency and meaning maps, but a lower image bias in actual memory (new Figure 5b). These findings serve as a meaningful launching point for creating better models of recall memory in the future (added Discussion on lines 533-549). Larger-scale studies with more images and more drawings per image will be needed to create these improved models.

Also, we previously aggregated saliency scores across pixels of an object by averaging them. However, **we have now conducted an additional analysis aggregating the scores by taking**

the peak score within an objects' pixels. We find that all statistical findings remain the same using this different aggregating method (lines 385-387). Finally, we have also added detail about the implementation of both GBVS and the Meaning Maps in the Methods (new section on Image-based Models).

Fourth, interrogating the contents of perceptual representations (including memory) with drawing tasks is hardly new, although the paper frames this approach as if it were, except for brief mention of work on boundary extension. Perhaps one of the most classic and iconic antecedents is Bartlett's experiments on serial reproduction of images from memory, which led to longstanding interest in how recall tasks can uncover subtle encoding biases, and/or lead to more schematized memory representations abstracted from the original sensory input. It might be useful to contextualize this work with respect to this theoretical backdrop, so the contribution of the current research can be more readily appreciated. Other work that may be relevant to review (and respond to) includes developmental studies on differential tendencies to draw what one knows ("intellectual realism," semantic knowledge) vs. what one sees ("visual realism," perceptual fidelity to source) across development -- e.g., studies by Paul Light, Gavin Bremner, and perhaps also Freeman & Janikoun (1972), Kosslyn et al. (1977). In addition, it might be useful to refer to Cohen & Bennett (1997), which asks which stages of processing most constrain accurate observational drawing (perception, decision-making, motor control, drawing evaluation). More recent psychophysical work of interest includes Florian Perdreau & Patrick Cavanagh's recent papers on effects of expertise on object encoding accuracy, as well as the work of Rebecca Chamberlain (w/ McManus and Wagemans).

Thank you for these references – they indeed explore very interesting ways to use drawing to get at both mental representations of objects as well as cognitive differences between drawing experts and non-experts. **We have added additional literature review reflecting drawing-related work to the Introduction of the paper** (lines 57-62).

Fifth, the stated contribution of this paper is methodological in nature (drawings can be used to measure visual long-term memory), rather than theoretical (using drawings to measure memory told us X,Y,Z new things about memory). In the introduction the authors suggest that such visual recall tasks could tell us something different from what we could learn from other visual tasks, e.g. recognition. Could the authors more explicitly state what we have learned about visual memory (rather than about the method) from using this visual recall paradigm? Without this, the theoretical significance of this paper is unclear. There are two ways that I believe that its significance could be substantively enhanced: (1) by including an additional memory-drawing condition that manipulates the delay either by shortening it (see Suggestions above) or extending it relative to the current main experiment, providing direct measurement of how the content and organization of information in visual long-term memory is affected by the relative demands on encoding vs. maintenance. (2) The other main way would be to strengthen the modeling section, motivating the use of different candidate visual encoding algorithms with respect to specific hypotheses about the role of perceptual, attentional mechanisms on constraining inputs to visual long-term memory, and comparing these different models' performance as a way of teasing apart these different hypotheses.

We have rewritten the Introduction and Discussion to be more explicit about the theoretical contributions from this work. Namely, beyond introducing a new method to assess visual recall memory, we think this work serves as an important exploration of what content exists within visual recall memory. Just as it is important to know the visual stimuli one shows participants in a visual experiment in order to interpret its results, it is important to understand the content of memory in order to fully understand memory-related phenomena. Verbal recall work has hypothesized very limited capacity of 5-9 objects (e.g., Miller, 1956), while visual

recognition work has demonstrated high capacity (Brady et al., 2008), but with low detail (Cunningham et al., 2015) and low spatial information (Standing, 1970). There is very limited work addressing the capacity and detail of the visual information we freely recall from memory, in spite of the fact that recognition and recall rely upon separate neural mechanisms (Holdstock et al., 2002; Staresina & Davachi, 2006). Our work thus provides an important and extensive characterization of the level of object and spatial detail within visual recall memory. When just looking at the number of images recalled, one may conclude participants can only recall about 12 items, however, we demonstrate that within those 12 items underlies a much higher capacity of memory, with on average 11 recalled objects per image, providing a new baseline estimate for visual recall capacity. We find extremely few false alarms, in spite of visual recognition work or verbal recall work that may have hypothesized many more errors (Deese, 1959; Simons & Rensink, 2005). We also find high spatial detail within recalled memories, for both location and object size. Such spatial detail has not yet been quantified by visual recognition tasks nor verbal recall tasks, and in fact participants often make spatial errors in recognition memory (Standing, 1970). We are not aware of other work that has shown this level of detail present in the content of visually recalled memories.

We have also followed the reviewer's suggestions of 1) adding an immediate recall (working memory) experiment and 2) looking at the contributions of different computational models of visual information to increase the theoretical impact of the current work. The Immediate Recall experiment serves to show that while the number of objects recalled decreases even after 11 minutes, the precision of the remembered objects remains (no change in false memories, or recalled object size). This also serves as an important point of comparison for future work which could look at visual recall decay over longer delays. The addition of the Meaning Maps and new semi-partial correlation analyses to look at unique explained variance also provides deeper insight into the possible mechanisms that may contribute to these results found in recall memory, but also reveals weaknesses in current models designed for attention and perception in modeling memory performance.

Minor: Could the authors comment on the decisions about drawing tools available, including the use of colored ink, in their task? It appears that several (but not all) drawings made use of color, but I was not able to find mention of this being evaluated systematically.

We have added text in the Methods (lines 695-697) to elaborate on the tools available and the use of color: "Participants were provided a black ballpoint pen and colored pencils and were instructed to optionally color or label aspects of the image. Ultimately 56.3% of drawings contained color." We did not analyze color for this experiment as it was optional for the participant, but color would be a useful metric to look at object detail for future studies.

Minor: The measurement of spatial error used is limited by its reliance on x,y displacement on the picture plane, but this measure is insensitive to errors involving rotation, scaling, and occlusion. Could the authors comment on the potential limitations of the current method for measuring spatial reconstruction error?

Following the reviewer's comment, **we extracted object scaling information from our current data.** For the spatial location analyses, AMT workers put ellipses around objects in the drawings,

enabling us to also analyze size. We have now added new analyses (lines 340-350) and graphs (Figure 4a) showing how object width and height vary across conditions. Interestingly, we find object width and height to be accurate and to show no difference across drawing conditions, regardless of memory. We have also added text referencing rotation and occlusion as important properties for future exploration (line 603).

Reviewers' comments:

Reviewer #1 (Remarks to the Author):

Since this is a revised manuscript I will not be providing a summary of the study as I see it. I believe that this revised manuscript is much improved and provides now much more valuable information. I especially like the comparison of objects recalled across different condition: immediate recall, delayed recall, category and image drawing. I think this analysis is very useful to help us start understanding more about what makes an object or image detail an idiosyncratic detail for memory. The additional experiments and analysis really work well in giving a more complete picture. Overall, I am happy with the responses to my original comments with some minor exceptions (see below).

1. I would suggest that the authors refrain from hyperbole and just report what they have using more measured language. For example: "Ultimately, we reveal an impressive amount of detail within visual recall memory, with observers creating precise drawings from memory for novel scene images, containing accurate object and spatial detail." I think there is no measure of impressive and it is up for debate what should be impressive. Also how impressive is it if the "Recall participants accurately recalled 12.1 images (out of 30) on average (SD = 4.0, MIN = 5, MAX = 20)" but "a similar free recall task with verbal stimuli, participants on average recalled 16.7 scene category names out of 30 (SD = 5.7; see Supplemental Information)" Furthermore, this is still an artificial situation since none of the participants remembering these images were under normal ecological conditions where their attention is being loaded like it is in real life situations. I feel that there is enough really interesting findings in the study without a need for an inflated punch line.

2. The authors now specify the details of the digit span task that would most certainly prevent verbal rehearsal though their argument for visual is really a stretch. There are studies that have tested the relationship between digit span and precision of visual working memory and find no correlation (just one for example: Zokaei, N., Burnett Heyes, S., Gorgoraptis, N., Budhdeo, S., & Husain, M. (2015). Working memory recall precision is a more sensitive index than span. *Journal of Neuropsychology*, 9(2), 319-329.). Thus, the digit span task used like the authors report, looking at capacity and not precision does not adequately tap into visual memory. However, it is interesting that the digit span task performance showed no correlation with either the visual or verbal free recall performance. This is interesting and should really be put forth more suggesting that the two systems can work independently, at least when they are sequentially engaged. It can also mean that this task did not do its job really if the idea was to prevent verbal or visual rehearsal.

3. Can the authors clarify this: "To test this, AMT workers (N = 24 per image, 1101 overall) were asked to match each drawing to one of three images from the same scene category." Does this mean that 24 workers judged all 30 drawings of one image or that 24 workers scored one drawing of an image?

4. I am a bit confused by the conclusion drawn from a non-significant results reported here;" There was no significant difference in matching between these cued recall drawings

and free recall drawings (non-parametric two-tailed independent samples Wilcoxon rank sum test: $Z \sim 0$, $p \sim 1$), indicating that there may be more images of equally high detail in memory than participants are able to initially retrieve." If there was no difference between cued and free-recall how can then there be a conclusion of actually there is more "equally high detail in memory than participants are able to initially retrieve." To me this means that actually, free recall gives you all there is in memory and cues do not provide you with anything more.

5. The authors responded to a previous comment with: " Since there has been no prior work using this type of paradigm on this scale, we unfortunately did not have data to use to perform a power analysis. However, we estimated that 30 participants should be sufficient to be able to capture meaningful effects in our data. Importantly, these data will now provide a basis for power analyses in similar future studies."

Which is fair enough but then they went and did additional studies in response to other reviewers (I applauded this) but not again doing a power analysis even that they now had the data for doing so. The three follow up studies had all different sample sizes (15, 24 and 30) that seem arbitrary which is really makes it seem like there was p hacking. I still maintain this is a problem for the paper and as such the claims have to be seen in this light.

6. I would like to point out to the authors that my comment of "If the authors would like to claim that massive visual long-term memory is very detailed than using only 30 images from 30 different semantic categories is not really probing that massive visual recognition memory." Does not imply that I am not aware that they are looking into detail rather than capacity. I also very much applaud this. My comment was more geared toward the language the initial manuscript used taunting results that I thought were not there. The impression I got from the first version of this manuscript was that it claimed visual long-term memory is not only massive but it is also detailed. I disagree and your additional experiment with immediate recall speaks to that. There is decay that is significant and that is only after 11 minutes.

7. A small point but important, from this sentence "Based on prior work, it might have been assumed that memory information would be imperfect, sparse, and frail [4,43,44]," a reader could be misled to think this is all of the work. The Brady et al, 2008 as well Hollingworth papers have already given evidence that this is not completely true of visual long-term memory. The current paper is another more elaborate and I find very useful evidence that that is not true. I think it would be negligent to claim that the Brady and other studies do not exist, and this sentence needs to better reflect the standing of current belief about the precision of visual long-term memory. It is not black and white.

Reviewer #2 (Remarks to the Author):

The authors have done a thorough job addressing my original concerns. I am happy to accept the manuscript as is.

Reviewer #3 (Remarks to the Author):

Overall, I believe this revised version of the manuscript to be a substantial improvement over the original submission. I appreciate the authors' general responsiveness to our concerns, questions, and suggestions. The inclusion of the Immediate Recall experiment was particularly useful for enhancing the paper's impact, and the revision communicates the results more clearly with less interpretational overextension. On the other hand, I found the theoretical contribution of including the alternative image saliency model to be less clear. Overall, upon carefully reviewing revised manuscript and the rebuttal letter, there are still a few concerns that I believe would need to be addressed before I could recommend this paper for publication.

For example, I still have serious concerns about the interpretability of the Category drawing condition as a primary benchmark for comparing with any of the drawing conditions. In particular, image-matching performance for a Category drawing does not really make sense when both of the distractors in the memory test array are also images from the same scene category. The authors acknowledge that there is no ground truth for this condition on this task, yet one of the most striking differences in Figure 1b is that between the category drawing bar and the other three, even though this emphasis is misplaced. Indeed the gray bar can only reflect chance guessing, so all this shows is that there is more information in image drawings than merely category-level information, but I am not sure if it supports the conclusion that "drawings from memory are *highly* diagnostic of the original image" -- the header for this section of the results. Does being merely more diagnostic than drawings never made from any image mean that the drawing is "highly" diagnostic? Drawings from long-term memory are indeed the least diagnostic of the original image of any of the image drawing conditions. Along the lines of both my and the other reviewers' comments from the initial round, I think tempering subjective language (perhaps editing from "Drawings from memory are *highly* diagnostic of the original image"  "Drawings from memory contain image-specific information") will make these results easier for readers to appreciate the content of the findings, with less distraction.

Another place where I might recommend reducing subjective language is in the final sentence of the abstract: "...but those memories contain impressively detailed representations of our visual experiences." Dropping the word "impressively" does little to detract from the scientific communication value of the abstract, I believe. I also do think some of the limitations of comparing image-based drawing task data to the category label drawing task data should be acknowledged in the Discussion.

line 275-279: "Category Drawing participants drew on average 58.4 (SD = 24.7) additional objects, which can be expected as Category Drawing participants were not drawing from a specific image. Category Drawings thus contained significantly more false objects than Delayed Recall drawings ($Z = 9.42$, $p = 4.24 \times 10^{-21}$)."

Related to the point above, the measurement of "false alarms" for category drawings does not really make sense in the context of the category drawing task -- false alarms are

undefined at the image level for the category drawing task, although they might be defined at the category-level (e.g., including an object that would be considered to not belong to a scene of that category). Given that this metric is problematic, I recommend that the reviewers revise this results section to reduce/remove the emphasis on the comparison of false alarm rate between the category drawing and the image/delayed/immediate image drawing conditions.

Both on my initial reading and during this round of review, I thought that the "number of objects" metric insufficiently defined/explained -- how did the experimenter determine what an object was when performing the initial segmentation using LabelMe? For objects that contain multiple component objects -- for example, a bed consisting of a mattress, frame, a pillow, blanket, etc. -- would drawing the bed count as a single object or multiple objects? Could this be elaborated upon in the Methods section? Otherwise, the key "number of objects" metric is much more difficult to interpret.

The pilot results from the 1-sentence description task are really intriguing. As a reminder, what motivated my initial suggestion to collect 1-sentence natural language descriptions (although more sentences might be more natural, and more comparable to the drawing task) of each scene was that this was yet another difference between the category-label-only drawing task and any of the drawings tasks (whether from memory or not). If the point is that drawings made from images contain image-specific information that you cannot easily get using conventional free recall methods --- it would be important to show that the kind of image-specific information you get in a detailed natural language description is less diagnostic or otherwise different somehow of the original image than a drawing of it. It sounds from the task description given in the rebuttal letter that the easiest way to address this would be to measure how well participants could identify the source image from which the natural language description was generated relative to exactly the same set of distractors. This would be much more informative as to the relative diagnosticity of the visual recall method vs. the most standard choice of non-visual recall method.

line 226-229: "While Delayed Recall drawings contained many objects beyond the Category Drawings, Delayed Recall drawings also often missed more common objects that would be present in a canonical representation of a specific scene type, but were less salient in the specific scene exemplar."

This is a super intriguing observation/possibility, and should be further probed with a quantitative measure that directly tests it. For instance, we should expect there to be greater variance in the type/position of objects across drawings from different images from same scene category than from category drawings. Moreover, we should expect that the type of objects that are present in the category drawings are also less likely to be recalled in drawings than the average object in that image. This could probably be computed from the heatmap representation, and would be really helpful for quantitatively evaluating this interesting intuition.

line 378: "We specifically tested two different algorithms shown to model image perception and be predictive of human fixation patterns: Graph-Based Visual Saliency (GBVS, [38]), and Meaning Maps [39]."

Currently not enough information/context is given about either of these models in this section for the reader to understand why they should be good predictive models of what people will include in their drawings. This section should be revised to better motivate each of these models, why they are relevant to test, what distinguishes them, and what each might or might not capture.

Regarding the relationship between recognition and recall, could the authors please also report the strength of the correlation between the number of objects recalled per image and the probability of recognition? One possibility is that even if the number of participants recalling an image is not correlated with the number who recognized it, that the strength of the memory for constituent components of the image might be related to recognition.

Thank you for the reviews of our manuscript NCOMMS-18-01825A, “Highly diagnostic and detailed content of visual memory revealed during free recall of real-world scenes”. We greatly appreciate the reviewers’ comments and have made substantial changes to address their remaining concerns. Specifically, we have:

1. **Conducted two new experiments (N=70 and N=364) to investigate the diagnosticity of verbal representations of an image (Reviewer 3).** The Results and Discussion have been modified with these new results, and a new section describing these experiments in detail has been added to the Supplemental Information.
2. **Implemented automatic part-of-speech and word lexicon tagging to analyze these verbal representations (Reviewer 3).** This allowed us to make objective quantifications of the 900 verbal descriptions from the new experiments. These results have been added to the Supplemental Information.
3. **Conducted new analyses on recalled object information (Reviewer 3).** We have conducted new analyses exploring the links between canonical and recalled objects, as well as image-based and recalled objects, and have updated the Results and Discussion accordingly. These analyses also resulted in a new section and two new figures in the Supplemental Information.
4. **Conducted additional analyses comparing recognition and object memory performance (Reviewer 3),** and we have updated the Results accordingly.
5. **Provided theoretical support and background for the image-based analyses (Reviewer 3).** We have added a thorough description to the Results.
6. **Elaborated on LabelMe object annotations (Reviewer 3).** We have added a new Methods section and Supplemental Information section on our use of the tool. Additionally, we will make all our annotations publicly available online, along with the code used in this experiment.
7. **Tempered subjective language throughout the text, lessened the emphasis on the Category Drawings condition, and elaborated on methods as asked by the Reviewers (Reviewers 1, 3)**

All changes have been marked in blue in the manuscript and Supplemental Information.

We think these substantial changes have greatly improved the manuscript.

Reviewer #1 (Remarks to the Author):

Since this is a revised manuscript I will not be providing a summary of the study as I see it. I believe that this revised manuscript is much improved and provides now much more valuable information. I especially like the comparison of objects recalled across different condition: immediate recall, delayed recall, category and image drawing. I think this analysis is very useful to help us start understanding more about what makes an object or image detail an idiosyncratic detail for memory. The additional experiments and analysis really work well in giving a more complete picture. Overall, I am happy with the responses to my original comments with some minor exceptions (see bellow).

We thank the reviewer for their thoughtful comments. We are very happy they found the additional experiments and analyses useful, and the new version much improved. We have made

further edits to the work to address the specific suggestions, including tempering our subjective language.

1. I would suggest that the authors refrain from hyperbole and just report what they have using more measured language. For example: "Ultimately, we reveal an impressive amount of detail within visual recall memory, with observers creating precise drawings from memory for novel scene images, containing accurate object and spatial detail." I think there is no measure of impressive and it is up for debate what should be impressive. Also how impressive is it if the "Recall participants accurately recalled 12.1 images (out of 30) on average (SD = 4.0, MIN = 5, MAX = 20)" but "a similar free recall task with verbal stimuli, participants on average recalled 16.7 scene category names out of 30 (SD = 5.7; see Supplemental Information)" Furthermore, this is still an artificial situation since none of the participants remembering these images were under normal ecological conditions where their attention is being loaded like it is in real life situations. I feel that there is enough rally interesting findings in the study without a need for an inflated punch line.

We have reworded much of our text throughout the manuscript to avoid any hyperbole – for example, the words “impressive” or “highly diagnostic” no longer appear throughout the whole manuscript.

*2. The authors now specify the details of the digit span task that would most certainly prevent verbal rehearsal though their argument for visual is really a stretch. There are studies that have tested the relationship between digit span and precision of visual working memory and find no correlation (just one for example: Zokaei, N., Burnett Heyes, S., Gorgoraptis, N., Budhdeo, S., & Husain, M. (2015). Working memory recall precision is a more sensitive index than span. *Journal of Neuropsychology*, 9(2), 319-329.). Thus, the digit span task used like the authors report, looking at capacity and not precision does not adequately tap into visual memory. However, it is interesting that the digit span task performance showed no correlation with either the visual or verbal free recall performance. This is interesting and should really be put forth more suggesting that the two systems can work independently, at least when they are sequentially engaged. It can also mean that this task did not do its job really if the idea was to prevent verbal or visual rehearsal.*

We have changed the text in the manuscript to eliminate any claims that the digit span task would prevent visual working memory, and to clarify that the task was mainly employed to introduce a delay between study and test phases and to limit verbal rehearsal (lines 97-99, 744-748). We have also added a brief summary of the digit span results (lines 101-103), with a reference to the Supplemental Information so readers can more clearly see this lack of correlation between digit span performance and recall and recognition performance.

3. Can the authors clarify this: "To test this, AMT workers (N = 24 per image, 1101 overall) were asked to match each drawing to one of three images from the same scene category." Does this mean that 24 workers judged all 30 drawings of one image or that 24 workers scored one drawing of an image?

We have edited the text to clarify that it is “N = 24 separate participants per image” (line 135), though note that Turk workers could participate in multiple trials if they desired; we have edited this in the Methods: “Each worker could complete as many trials as they desired, and each worker on average completed 58.2 trials” (lines 834-835).

4. I am a bit confused by the conclusion drawn form a non-significant results reported here;" There was no significant difference in matching between these cued recall drawings and free recall drawings (non-parametric

two-tailed independent samples Wilcoxon rank sum test: $Z \sim 0$, $p \sim 1$), indicating that there may be more images of equally high detail in memory than participants are able to initially retrieve.” If there was no difference between cued and free-recall how can then there be a conclusion of actually there is more “equally high detail in memory than participants are able to initially retrieve.” To me this means that actually, free recall gives you all there is in memory and cues do not provide you with anything more.

We have edited the text to be clearer (lines 147-150). Following the free recall period, we conducted an additional recall phase in which participants were given an object name cue for each studied image and were asked to draw any additional images they now remembered. On average, participants drew an additional 5.7 images (SD = 2.9) that they didn't originally draw during free recall. We then conducted the same drawing matching task to investigate the diagnosticity of these new cued drawings and found they were just as diagnostic of their original image as drawings that were freely recalled ($Z \sim 0$, $p \sim 1$). Thus, with a cue, participants were able to draw additional images with a similar level of detail. This suggests that there may be more images in participants' memories than they originally recalled during the free recall phase.

5. The authors responded to a previous comment with: “Since there has been no prior work using this type of paradigm on this scale, we unfortunately did not have data to use to perform a power analysis. However, we estimated that 30 participants should be sufficient to be able to capture meaningful effects in our data. Importantly, these data will now provide a basis for power analyses in similar future studies.”

Which is fair enough but then they went and did additional studies in response to other reviewers (I applauded this) but not again doing a power analysis even that they now had the data for doing so. The three follow up studies had all different sample sizes (15, 24 and 30) that seem arbitrary which is really makes it seem like there was p hacking. I still maintain this is a problem for the paper and as such the claims have to be seen in this light.

We apologize if this was unclear. The studies with 15 and 24 participants were conducted simultaneously with the original experiment (not after the reviews), and the participant numbers were selected to match the number of samples in the main Delayed Recall experiment. Specifically, each image in the Delayed Recall experiment had 15 observations (there were 30 participants who saw 1 of 2 possible images per category), so 15 observations was used in other experiments to match this number. The maximum number of people who recalled any given image was 12 out of 15, so 12 observations were needed per image for the Image Drawing Experiment (or 24 participants total, given the 2 image sets). We have now clarified the reasoning behind our participant numbers in the Methods (lines 783-784, 796-797, 806-808, 816-817).

6. I would like to point out to the authors that my comment of “If the authors would like to claim that massive visual long-term memory is very detailed than using only 30 images from 30 different semantic categories is not really probing that massive visual recognition memory.” Does not imply that I am not aware that they are looking into detail rather than capacity. I also very much applaud this. My comment was more geared toward the language the initial manuscript used taunting results that I thought were not there. The impression I got from the first version of this manuscript was that it claimed visual long-term memory is not only massive but it is also detailed. I disagree and your additional experiment with immediate recall speaks to that. There is decay that is significant and that is only after 11 minutes.

We agree that we were unclear in the first version of the manuscript. We hope the current version is clearer about the nuances of the amount of detail in memory and that it decays over time.

7. A small point but important, from this sentence “Based on prior work, it might have been assumed that memory information would be imperfect, sparse, and frail [4,43,44],”, a reader could be misled to think this is all of the work. The Brady et al, 2008 as well Hollingworth papers have already given evidence that this is not completely true of visual long-term memory. The current paper is another more elaborate and I find very useful evidence that that is not true. I think it would be negligent to claim that the Brady and other studies do not exist, and this sentence needs to better reflect the standing of current belief about the precision of visual long-term memory. It is not black and white.

We have reworded the sentence to show both sides of the debate about the capacity and detail in memory (lines 553-555): “While prior work has posited large capacity to visual recognition memory (Landman et al., 2003; Hollingworth et al., 2004; Brady et al., 2008), other work has suggested recognition and recall memory may have surprisingly low detail (Cunningham et al., 2015; Levin & Simons, 1997; Loftus, 2005).”

Reviewer #2 (Remarks to the Author):

The authors have done a thorough job addressing my original concerns. I am happy to accept the manuscript as is.

Reviewer #3 (Remarks to the Author):

Overall, I believe this revised version of the manuscript to be a substantial improvement over the original submission. I appreciate the authors' general responsiveness to our concerns, questions, and suggestions. The inclusion of the Immediate Recall experiment was particularly useful for enhancing the paper's impact, and the revision communicates the results more clearly with less interpretational overextension. On the other hand, I found the theoretical contribution of including the alternative image saliency model to be less clear. Overall, upon carefully reviewing revised manuscript and the rebuttal letter, there are still a few concerns that I believe would need to be addressed before I could recommend this paper for publication.

The reviewer's comments were indeed very helpful for improving the previous version of the manuscript, and we are glad the impact of the paper has improved.

*For example, I still have serious concerns about the interpretability of the Category drawing condition as a primary benchmark for comparing with any of the drawing conditions. In particular, image-matching performance for a Category drawing does not really make sense when both of the distractors in the memory test array are also images from the same scene category. The authors acknowledge that there is no ground truth for this condition on this task, yet one of the most striking differences in Figure 1b is that between the category drawing bar and the other three, even though this emphasis is misplaced. Indeed the gray bar can only reflect chance guessing, so all this shows is that there is more information in image drawings than merely category-level information, but I am not sure if it supports the conclusion that "drawings from memory are *highly* diagnostic of the original image" -- the header for this section of the results. Does being merely more diagnostic than drawings never made from any image mean that the drawing is "highly" diagnostic? Drawings from long-term memory are indeed the least diagnostic of the original image of any of the image drawing conditions. Along the lines of both my and the other reviewers' comments from the initial round, I think tempering subjective language (perhaps editing from "Drawings from memory are *highly* diagnostic of the original image"  "Drawings from memory contain image-specific information") will make these results easier for readers to appreciate the content of the findings, with less distraction.*

We appreciate the reviewer's concerns about the Category Drawing condition and now realize we need to be much clearer about what the purpose of this condition is. The overall goal of the study was to reveal the visual content of memory, but we can't assume the Recall Drawings reflect such visual content. The Category Drawing condition is a way of estimating what performance would be like if there were no specific memory for the content of the image, just the overall identity or category of an image. We agree with the reviewer that in the diagnosticity test (which is only the first of the tests we conducted), performance for the Category Drawings primarily reflects chance guessing, but importantly it also allows us to take into account any potential biases in our stimuli (e.g. if some of our images happened to be better exemplars of that category than others). Indeed, there is a wide spread to the matching performance for the Category Drawings (Figure 1b), showing that chance-level matching is in fact not always the case. However, if we compare the same category across conditions in Figure 1 (the gray lines), we see the other drawing conditions are still more diagnostic than these Category Drawings. At the same time, the Category Drawings also allow us to look at how memory compares to a prototypical internal representation of that same category of scene, which is critical for analyses of object content. Thus, we consider the Category Drawings a critical test and have now made these points explicitly in the manuscript (lines 163-168, 171-174, 196-201). However, we also acknowledge that we shouldn't over-emphasize the comparison of the Category Drawings with the Recall Drawings and have also toned down our language.

We agree with the reviewer that the use of the word "highly" is potentially misleading in the context of the diagnosticity test and have tempered our language accordingly. We have also changed the title to remove references to diagnosticity. We think it is also important to note that in terms of our more general conclusions about the content in memory, the diagnosticity test was only the initial starting point and the subsequent tests (e.g. object content, spatial information) provide stronger metrics for characterizing the content of memory.

Another place where I might recommend reducing subjective language is in the final sentence of the abstract: "...but those memories contain impressively detailed representations of our visual experiences." Dropping the word "impressively" does little to detract from the scientific communication value of the abstract, I believe. I also do think some of the limitations of comparing image-based drawing task data to the category label drawing task data should be acknowledged in the Discussion.

We agree and have tempered our subjective language; we have removed "impressively" from the abstract, as well as the word "impressive" throughout the rest of the manuscript. We have also added text to the Discussion (in addition to the Results section) to clarify the role of the Category Drawing task (lines 163-168, 171-174, 196-201, 652-659).

line 275-279: "Category Drawing participants drew on average 58.4 (SD = 24.7) additional objects, which can be expected as Category Drawing participants were not drawing from a specific image. Category Drawings thus contained significantly more false objects than Delayed Recall drawings (Z = 9.42, p = 4.24 × 10⁻²¹)."
Related to the point above, the measurement of "false alarms" for category drawings does not really make sense in the context of the category drawing task -- false alarms are undefined at the image level for the category drawing task, although they might be defined at the category-level (e.g., including an object that would be considered to not belong to a scene of that category). Given that this metric is problematic, I recommend that the reviewers revise this results section to reduce/remove the emphasis on the comparison of false alarm rate between the category drawing and the image/delayed/immediate image drawing conditions.

We appreciate the reviewer's concern and now realize that the use of the term "false alarms" in the context of the Category Drawings is potentially misleading. We have updated the language to make it clear that these analyses are more about revealing potential additional objects that are also consistent with the category. The accuracy of memory is reflected not only in what the participants do draw, but also what they don't draw compared to the Category Drawings. For example, it could be possible that participants were recalling accurate object details, but also filling in many additional objects based on their prototypical internal representation of that scene category, in the same way that people often have false alarms during verbal recall of words from a similar semantic category (Roediger & McDermott, 1995. Creating false memories: Remembering words not presented in lists. *JEP: L, M, C.*). The analysis of the object content of the Category Drawings shows that there are many category-consistent objects that are not drawn from memory, suggesting that the extent to which participants are using semantic information to fill-in the drawings is limited and that they are maintaining an accurate memory of the image content.

Both on my initial reading and during this round of review, I thought that the "number of objects" metric insufficiently defined/explained -- how did the experimenter determine what an object was when performing the initial segmentation using LabelMe? For objects that contain multiple component objects -- for example, a bed consisting of a mattress, frame, a pillow, blanket, etc. -- would drawing the bed count as a single object or multiple objects? Could this be elaborated upon in the Methods section? Otherwise, the key "number of objects" metric is much more difficult to interpret.

We have added a paragraph in the Methods about how objects were defined in LabelMe and how many objects per image this resulted in (lines 719-730), added a citation on guidelines on how other annotators have decided what is an object or not [48], and added a section on example segmentations in the Supplemental Information (page 11). We will also make all of our LabelMe annotations available online with the rest of the data. For the bed example, only distinct, separable, and visible components were annotated – for example, here is a bedroom used in our experiment (next page):

Here, all pillows, the comforter, and the posts of the bedframe were annotated. However, since a mattress and thin sheets weren't visible (even though they probably were there), they were not annotated.

The pilot results from the 1-sentence description task are really intriguing. As a reminder, what motivated my initial suggestion to collect 1-sentence natural language descriptions (although more sentences might be more natural, and more comparable to the drawing task) of each scene was that this was yet another difference between the category-label-only drawing task and any of the drawings tasks (whether from memory or not). If the point is that drawings made from images contain image-specific information that you cannot easily get using conventional free recall methods --- it would be important to show that the kind of image-specific information you get in a detailed natural language description is less diagnostic or otherwise different somehow of the original image than a drawing of it. It sounds from the task description given in the rebuttal letter that the easiest way to address this would be to measure how well participants could identify the source image from which the natural language description was generated relative to exactly the same set of distractors. This would be much more informative as to the relative diagnosticity of the visual recall method vs. the most standard choice of non-visual recall method.

With the natural language descriptions the question is not so much about verbal recall versus drawing-based visual recall as tasks, but rather to investigate whether the Recall Drawings could reflect a verbal rather than a visual memory. While the Category Drawings allow us to estimate what might be drawn in the absence of any memory for specific image content beyond the category label, it is still possible (as the reviewer noted in the previous round of review) that participants could be maintaining a detailed semantic or verbal memory representation that extends beyond the category label. We thus asked participants to generate one sentence descriptions of the images that they might use to help them remember the image because that seemed a reasonable length in the context of trying to remember 30 images. To analyze the content of these verbal descriptions, we have now extended this experiment and re-analyzed the data using more objective criteria (lines 196-205, 652-659 in the main manuscript and pages 14-17 in the Supplemental Information).

We collected an additional 5 verbal descriptions per image (to have 15 participants per image, the same number of participants as those who studied the images for the drawing tasks). We then performed two new analyses on these descriptions: 1) online participants (N=364) matched the descriptions to their images (*test of diagnosticity*) – analogous to the diagnosticity test for the drawings, 2) automatically analyzed the content of the descriptions using a part-of-speech tagger and two word lexicons (*test of detail*).

In terms of diagnosticity, we find that the verbal descriptions are matched to their corresponding image more frequently than Category Drawings. These descriptions are matched correctly less frequently than drawings made from the images, and with equal frequency to the Delayed Recall drawings. Thus, the verbal descriptions are diagnostic of their image, and may have similar diagnosticity to a memory representation (though note the drawings were made during recall for multiple images, while the descriptions were made for single images with no memory burden). However, as we discussed earlier, diagnosticity is only an initial measure of what content is within a representation and can be determined by a few salient features or objects. Thus, it is critical to evaluate the content of the verbal descriptions.

To analyze the content of the descriptions, we quantified the sentences in three ways: 1) counted number of concrete nouns to estimate object detail (using the word concreteness database by Brysbaert et al., 2014), 2) counted number of spatial signal words to estimate spatial detail (Fry et al., 1993), and 3) counted number of adjectives and adverbs to estimate subjective detail. All of these quantifications were run automatically by a part-of-speech tagger (Toutanova et al., 2003). We find that verbal descriptions contain low amounts of detail, with only 2.8 objects and 1.1 spatial words on average per sentence. This shows that any such verbal descriptions used to encode an image from memory would not be able to explain the levels of object detail and spatial precision (in terms of both location and size) that we see in the drawings. Of course, more detailed verbal descriptions could be generated for these images (e.g., allowing more than one sentence), but there is the larger question of what amount of verbal detail could be reasonably held in memory for 30 images. We think it unlikely that verbal memories could contain such spatial detail, but we cannot rule out its contribution. We also find that these sentences contain various subjective details, with 1.7 adjectives and adverbs (e.g., *beautiful*, *messily*) on average, and several of the sentences are highly idiosyncratic (e.g., “My dream vacation pool.”). This may point towards the idea that these verbal descriptions could be serving as cues to other familiar memories. Ultimately, it could be that the memories for the images in our current study are a combination of both visual details as well as verbal details, and such a question of how visual and verbal strategies interact to form a memory would be highly intriguing for future work.

line 226-229: "While Delayed Recall drawings contained many objects beyond the Category Drawings, Delayed Recall drawings also often missed more common objects that would be present in a canonical representation of a specific scene type, but were less salient in the specific scene exemplar."

This is a super intriguing observation/possibility, and should be further probed with a quantitative measure that directly tests it. For instance, we should expect there to be greater variance in the type/position of objects across drawings from different images from same scene category than from category drawings. Moreover, we should expect that the type of objects that are present in the category drawings are also less likely to be recalled in drawings than the average object in that image. This could probably be computed from the heatmap representation, and would be really helpful for quantitatively evaluating this interesting intuition.

We have probed this observation further with additional analyses (Supplemental Information pages 18-20). While we cannot directly contrast the type and position of objects in the Delayed Recall condition versus the Category Drawing condition because of the difference in exemplar numbers (i.e., there are 15 individual Category Drawings per category, but recall drawings are only made from 2 images of the category), we can compare the likelihood of recalling a given object to the likelihood of drawing it given its category name. Supplemental Figure 3 (copied here) shows the proportion of times a given object was present in the Delayed Recall drawings compared to the Category drawings. There are objects in all quadrants of the plot; there are objects that are drawn in both conditions (e.g., bed in a bedroom) as well as objects that are drawn in neither (e.g., steps in a fountain scene). There are also objects that are drawn more frequently during Delayed Recall than during Category Drawing (e.g., cannons in a castle), as well as objects drawn more frequently during Category Drawing than Delayed Recall (e.g., the sofa in the corner of a living room). These results show that participants are successfully recalling many objects that are non-canonical for a scene, but they also fail to recall some canonical objects.

Supplemental Figure 3 – (Left) Scatterplot of all objects in the experimental images, showing the rate at which they are drawn in Category Drawings versus Delayed Recall Drawings. Red dots indicate objects that were drawn more often from Delayed Recall than in Category Drawings, while blue dots indicate objects more often drawn in Category Drawings. Black dots indicate objects drawn with equal frequency. The histograms indicate the number of objects with a given drawing rate, sorted by condition. Highlighted points indicate example objects (at the right), bordered with the same corresponding color, to show example objects at the extreme ends of both axes.

We have also performed a similar analysis comparing the frequency of recall with the frequency of drawing an object directly from the image (see Supplemental Figure 4 below from the Supplemental Information). Interestingly, we again see a spread of objects; while more commonly participants are failing to recall an object that would be drawn given an image, there are also objects that are drawn more frequently from memory than from perception. This will serve as an interesting topic for future work – what is happening to these objects to make them more salient in memory than perception?

Supplemental Figure 4 – (Left) Scatterplot of all objects in the experimental images, showing the rate at which they are drawn in Image Drawings versus Delayed Recall Drawings. Red dots indicate objects that were drawn more often from Delayed Recall than in Image Drawings, while blue dots indicate objects more often drawn in Image Drawings. Black dots indicate objects drawn with equal frequency. The histograms indicate the number of objects with a given drawing rate, sorted by condition. Highlighted points indicate example objects (at the right), bordered with the same corresponding color, to show example objects at the extreme ends of both axes.

line 378: "We specifically tested two different algorithms shown to model image perception and be predictive of human fixation patterns: Graph-Based Visual Saliency (GBVS, [38]), and Meaning Maps [39]."
 Currently not enough information/context is given about either of these models in this section for the reader to understand why they should be good predictive models of what people will include in their drawings. This section should be revised to better motivate each of these models, why they are relevant to test, what distinguishes them, and what each might or might not capture.

We have added several sentences (lines 399-417) to strengthen the motivation behind using these models, and described their different features.

Regarding the relationship between recognition and recall, could the authors please also report the strength of the correlation between the number of objects recalled per image and the probability of recognition? One possibility is that even if the number of participants recalling an image is not correlated with the number who recognized it, that the strength of the memory for constituent components of the image might be related to recognition.

This is an interesting point. We have conducted this additional analysis and found no correlation between number of objects recalled per image and recognition rate ($\rho = -0.02$, $p = 0.867$; lines 516-519). This shows that recognition is also not necessarily linked to the amount of detail (or memory strength) for recalled images.

REVIEWERS' COMMENTS:

Reviewer #1 (Remarks to the Author):

The authors have done a thorough job addressing my original concerns. I am happy to accept the manuscript as is.

Reviewer #3 (Remarks to the Author):

Overall, I believe this revised version of the manuscript to be greatly improved over the previous revision. I am particularly impressed by the authors' proactive responses to our concerns, questions, and suggestions. The enhanced clarity/precision of the writing and the inclusion of additional experiments and analyses (e.g., of the content of Category vs. Recall drawings, of the verbal descriptions) have served to further enhance the impact of the paper, and I believe as a whole this paper will provide a valuable contribution to our understanding of visual memory.

(1) Regarding this line in the Introduction:

"However, largely due to the complexity and subjectivity of drawings, such studies have often used small stimulus sets with simple metrics of interest (e.g., [24]) or subjective experimenter ratings (e.g., [10,29,30]), without delving into the rich content within these drawings."

There is some relevant work on using deep CNNs to measure content in drawings that should probably be cited here, by Fan et al in CogSci:
<https://onlinelibrary.wiley.com/doi/abs/10.1111/cogs.12676>. Not looking at visual memory, but definitely relevant to the questions raised by this study. Some additional language should also be added to the Discussion to mention that a key challenge for future work is to enhance modern vision models with the capacity to represent the content differences between Recall/Image/Category drawings in an appropriate way, and the crowdsourced data/annotations here could help validate those approaches.

(2) Regarding the Verbal Description control, the new quantitative results from this control are actually really interesting, and I have two concrete suggestions: (i) I think including a modification of the bar graph in Figure 1 to include matching accuracy from the verbal description would be useful to include in Supplemental Figure 2 (at the very least), so that readers can see the quantitative comparison between the verbal description condition and the others in context. (ii) I think that adding language in the Discussion that explicitly recommends that future work investigate differences in linguistic vs. graphical representations produced from memory w.r.t. diagnosticity, concrete detail. How would a matched language based version of this visual recall task actually compare in terms of the content produced, if it were allowed to be as free-form as in the current study (permitting extended descriptions in a free recall setting). It is also not clear to me that the verbal descriptions were generated under the same goal (to be as detailed as possible) as the drawings were, so I think that this gap should be explicitly stated as a direction for future

research.

This passage on lines 657-659 goes some way towards posing this question:

"Further experimentation will be needed to directly analyze and compare the amount of detail within verbal and visual representations of a scene, and see to what degree verbal and visual strategies may work in conjunction to form a memory."

...but I think this could be done in a more explicit manner. Right now, the emphasis is solely placed on a hypothesis about the "strategies" people may be using to encode the scene in a verbal vs. visual format. In addition, I think that understanding the potentially different affordances of these two output modalities (verbal, visual) is equally important for gaining a full understanding of the contents of memory, and putting them on an even playing field will help resolve similarities and differences between these two modalities more clearly. E.g., the relative lack of spatial and concrete detail in verbal descriptions would be *much* more compelling and interesting if they were generated under exactly the same instructions/goals as the drawings had been.

We greatly appreciate these new comments on our manuscript. We have addressed Reviewer #3's latest set of comments, and note our changes in blue below.

Reviewer #3 (Remarks to the Author):

Overall, I believe this revised version of the manuscript to be greatly improved over the previous revision. I am particularly impressed by the authors' proactive responses to our concerns, questions, and suggestions. The enhanced clarity/precision of the writing and the inclusion of additional experiments and analyses (e.g., of the content of Category vs. Recall drawings, of the verbal descriptions) have served to further enhance the impact of the paper, and I believe as a whole this paper will provide a valuable contribution to our understanding of visual memory.

We are happy that the Reviewer is satisfied with the improvements to the paper.

(1) Regarding this line in the Introduction:

"However, largely due to the complexity and subjectivity of drawings, such studies have often used small stimulus sets with simple metrics of interest (e.g., [24]) or subjective experimenter ratings (e.g., [10,29,30]), without delving into the rich content within these drawings."

There is some relevant work on using deep CNNs to measure content in drawings that should probably be cited here, by Fan et al in CogSci: <https://onlinelibrary.wiley.com/doi/abs/10.1111/cogs.12676>. Not looking at visual memory, but definitely relevant to the questions raised by this study. Some additional language should also be added to the Discussion to mention that a key challenge for future work is to enhance modern vision models with the capacity to represent the content differences between Recall/Image/Category drawings in an appropriate way, and the crowdsourced data/annotations here could help validate those approaches.

We have added a description and citation in the Introduction for Fan et al.'s work, as well as related work by Eitz et al. (lines 65-66). We have also added text in the Discussion mentioning how the current paradigm will enhance modern vision models (lines 613-616).

(2) Regarding the Verbal Description control, the new quantitative results from this control are actually really interesting, and I have two concrete suggestions: (i) I think including a modification of the bar graph in Figure 1 to include matching accuracy from the verbal description would be useful to include in Supplemental Figure 2 (at the very least), so that readers can see the quantitative comparison between the verbal description condition and the others in context.

As suggested, we have added a new Supplementary Figure 4 with a modified bar graph including matching accuracy from the verbal description task.

(ii) I think that adding language in the Discussion that explicitly recommends that future work investigate differences in linguistic vs. graphical representations produced from memory w.r.t. diagnosticity, concrete detail. How would a matched language based version of this visual recall

task actually compare in terms of the content produced, if it were allowed to be as free-form as in the current study (permitting extended descriptions in a free recall setting). It is also not clear to me that the verbal

descriptions were generated under the same goal (to be as detailed as possible) as the drawings were, so I think that this gap should be explicitly stated as a direction for future research.

This passage on lines 657-659 goes some way towards posing this question:

"Further experimentation will be needed to directly analyze and compare the amount of detail within verbal and visual representations of a scene, and see to what degree verbal and visual strategies may work in conjunction to form a memory."

...but I think this could be done in a more explicit manner. Right now, the emphasis is solely placed on a hypothesis about the "strategies" people may be using to encode the scene in a verbal vs. visual format. In addition, I think that understanding the potentially different affordances of these two output modalities (verbal, visual) is equally important for gaining a full understanding of the contents of memory, and putting them on an even playing field will help resolve similarities and differences between these two modalities more clearly. E.g., the relative lack of spatial and concrete detail in verbal descriptions would be **much** more compelling and interesting if they were generated under exactly the same instructions/goals as the drawings had been.

We have edited lines 657-659 and added a new sentence (now lines 684-691) explicitly stating the need for such experiments putting visual and verbal output tasks on a level playing field, to really examine the differences in memory content detail.